# Interacting edge states of fermionic symmetry-protected topological phases in two dimensions

**Joseph Sullivan and Meng Cheng⋆**

Department of Physics, Yale University, New Haven, CT 06511-8499, USA

⋆ m.cheng@yale.edu

## Abstract

Recently, it has been found that there exist symmetry-protected topological phases of fermions, which have no realizations in non-interacting fermionic systems or bosonic models. We study the edge states of such an intrinsically interacting fermionic SPT phase in two spatial dimensions, protected by $\mathbb{Z}_4 \times \mathbb{Z}_2^\mathsf{T}$ symmetry. We model the edge Hilbert space by replacing the internal $\mathbb{Z}_4$ symmetry with a spatial translation symmetry, and design an exactly solvable Hamiltonian for the edge model. We show that at low-energy the edge can be described by a two-component Luttinger liquid, with nontrivial symmetry transformations that can only be realized in strongly interacting systems. We further demonstrate the symmetry-protected gaplessness under various perturbations, and the bulk-edge correspondence in the theory.



# 1   Introduction

Symmetry-protected topological (SPT) phases [1–3] are characterized by their protected boundary states. The protecting symmetries act anomalously on the boundary states, in such a way that a symmetric and non-degenerate ground state is prohibited. As a result, a boundary without symmetry breaking must be gapless, or gapped with intrinsic topological order when the boundary is two-dimensional or higher [4]. Many examples of SPT phases have been discovered in fermionic systems, in particular in electronic band insulators and BCS superconductors [5–8]. Their gapless boundary states are well-understood thanks to the non-interacting nature of the states and can be described in terms of Dirac or Majorana fermions when the interactions on the boundary are sufficiently weak. Dirac-like surface states have been observed in 3D time-reversal-invariant topological insulators [7,8], as well as their generalizations with crystalline symmetries. Strong interactions can drive the gapless surface to symmetry-enriched topologically ordered phases [4,9–13].

Beyond free fermions, recently there has been significant theoretical progress in classifying SPT phases in interacting fermionic systems [14–19], following previous classifications of bosonic SPT phases using group cohomology [1]. A number of different approaches have been put forward, such as fermionic generalizations of the group-cohomology constructions [14, 18,20], and classifications based on topological quantum field theories [21–24]. These results have pointed to an interesting possibility, namely interacting Fermionic Symmery-Protected Topological (FSPT) phases, which can only exist with strong interactions. One mechanism for such phases is when fermions first form bosonic molecules/spins under strong interactions, and then these bosons form a SPT state. To give an example, imagine in 2D fermions first form charge-$2e$ bosons, and then the bosons are put into a so-called bosonic integer quantum Hall state [25], which has an electric Hall conductance quantized to $\frac{8e^2}{h}$, but a vanishing ther-

mall Hall conductance, violating the Wiedemann-Franz law [26]. Thus this phase can only be found in the presence of strong interactions. However, more interestingly there exist *intrinsically* fermionic phases which can not be realized by weakly interacting systems and have no bosonic counterpart. Examples of such intrinsically FSPT phases have been discovered in one, two and three dimensions [15, 17, 27]. In one dimension, an intrinsically interacting FSPT phase exists when the symmetry group is $\mathbb{Z}_4^f \times \mathbb{Z}_4$ [15, 27], where the edge modes transform as a projective representation of the symmetry group. Here $\mathbb{Z}_4^f$ refers to the conservation of fermion number mod 4. In two dimensions, the simplest symmetry group that allows an interacting FSPT phase is $\mathbb{Z}_2^f \times \mathbb{Z}_4 \times \mathbb{Z}_2^\mathsf{T}$. Here $\mathbb{Z}_2^\mathsf{T}$ denotes the time-reversal symmetry that squares to the identity, i.e. fermions are Kramers singlets. Similar states protected by crystalline symmetries have been found [28, 29].

Given that these new phases require strong interactions to exist, their boundary states can not be simply free Dirac/Majorana fermions. While exactly-solvable bulk Hamiltonians can in principle be constructed [14, 27], it is very desirable to have a physical understanding of the interacting edge states. Generally, nontrivial dynamics on the edge leads to either gapped phases with broken symmetry, or a symmetric gapless phase. We will address this question for the 2D $\mathbb{Z}_2^f \times \mathbb{Z}_4 \times \mathbb{Z}_2^\mathsf{T}$ FSPT phase. Our strategy is to study a closely related 2D crystalline FSPT phase, where the $\mathbb{Z}_4$ symmetry is replaced by a $\mathbb{Z}$ translation symmetry. The corresponding crystalline SPT phase has a simple bulk wavefunction, and the edge modes can be cleanly separated from the bulk as a stand-alone 1D chain of spinless fermions, which do not allow any quadratic couplings respecting the symmetries. We design an analytically solvable model for the boundary chain, and derive a two-component Luttinger liquid theory that captures the low-energy physics, based on which we propose a very similar theory where the spatial translation $\mathbb{Z}$ is replaced by an internal $\mathbb{Z}_4$ symmetry. We then demonstrate that the theory exhibits the correct quantum anomaly.

## 2 Intrinsically interacting FSPT phases in 2D

We review the physics of intrinsically interacting FSPT phases in 2D, through a decorated domain wall picture [30], and closely related ones with crystalline symmetries. Another construction of FSPT phases using group super-cohomology theory will be briefly summarized in Appendix A. We will focus on $G = \mathbb{Z}_4 \times \mathbb{Z}_2^\mathsf{T}$. Notice that fermions transform as Kramers singlet, e.g. spinless fermions, unlike the spin-1/2 electrons which are Kramers doublets.

We first briefly recall the non-interacting classification with such a symmetry group [31]. Since there is no charge conservation, in general the BdG Hamiltonian can be compactly written as $H = \Psi^\dagger h \Psi$, where the Nambu spinor $\Psi$ is schematically defined as $\Psi = (c, c^\dagger)$ suppressing all the indices (site, spin, etc.). In the presence of a unitary symmetry, e.g. $\mathbb{Z}_4$ in this case, the first-quantized Hamiltonian $h$ can be block diagonalized, with blocks labeled by $\mathbb{Z}_4$ eigenvalues. Now within each block the only symmetry is the $\mathbb{Z}_2^\mathsf{T}$. Because the fermions are Kramers singlets, the classification for each block is given by the BDI class in the ten-fold way, which is completely trivial in 2D. We conclude that the overall classification is trivial as well. Therefore strong interactions are necessary to form any nontrivial FSPT phases with this symmetry.

### 2.1 Decorated domain wall construction

The ground state wavefunction of many SPT phases can be understood through a decorated domain wall construction. One first imagines that a discrete symmetry $H$ is broken spontaneously. We take this discrete symmetry to be a normal subgroup of the protecting symmetry

of the SPT phase. Once the symmetry is broken, there can be domain walls between different symmetry-breaking patterns, e.g. different expectation values of an order parameter. Mathematically, each domain wall is uniquely labeled by a group element $\mathbf{h} \in H$. This process can be reversed: starting from the broken symmetry state, the symmetry can be restored by proliferating domain walls. In other words, the wavefunction of a symmetric state can be viewed as the quantum superposition of all possible domain wall configurations.

Now imagine that the domain walls are "decorated" by 1D SPT states protected by the remaining symmetry $G/H$. The decoration is in fact the manifestation of the SPT order in the symmetry-breaking phase, and can be understood more intuitively in the presence of a physical edge: while domain walls are closed in the bulk, they can end on the edge, which also terminate the associated 1D SPT states on the domain walls. Thus topologically protected zero-energy modes must appear at a domain wall on the edge. A SPT wavefunction is then obtained by proliferating domain walls decorated by 1D SPT states. Importantly, a consistent symmetric wavefunction requires that the decorated 1D SPT states obey the same group multiplication law as the domain walls. Namely, two domain walls labeled by group elements $\mathbf{h}_1$ and $\mathbf{h}_2$ can fuse into a domain wall labeled by $\mathbf{h}_1 \mathbf{h}_2$. The same relation must be satisfied by the associated 1D SPT states.

To illustrate, let us consider the example of class DIII topological superconductor (TSC) in 2D. As a simple model for DIII TSC, consider spin-1/2 electrons with $p_x + i p_y / p_x - i p_y$ pairing for spin up/down fermions. Time-reversal symmetry acts on fermion annihilation operators as $\psi_\alpha \to \sum_\beta (i\sigma^y)_{\alpha\beta} \psi_\beta$, where $\alpha, \beta = \uparrow, \downarrow$ are the spin $z$ component. Edge states of this TSC are described by helical Majorana fermions, where left- and right-moving modes carry opposite spins. They can be gapped out by turning on a time-reversal breaking mass term. Thus a time-reversal domain wall on the edge corresponds to a mass that changes sign. It is well-known that a Majorana zero-energy bound state is found at the mass domain wall. In the bulk, a domain wall then carries a Majorana chain which gives rise to the Majorana zero mode when it is cut open by the physical edge [1]. Hence the DIII TSC can be thought of as proliferating domain walls decorated by Majorana chains. Such a picture was realized recently in a commuting-projector model for the class DIII TSC [32].

For $G = \mathbb{Z}_4 \times \mathbb{Z}_2^T$ in 2D, we may write the wavefunction as a superposition of $\mathbb{Z}_4$ domain walls, and decorate them with 1D SPT states protected by the remaining $\mathbb{Z}_2^T$ symmetry. Denote the generator of the $\mathbb{Z}_4$ group by $\mathbf{g}$. The classification of 1D FSPT phases with $\mathbb{Z}_2^T$ symmetry is well-understood: non-interacting fermions with this symmetry fall into the class BDI in the periodic table, with a $\mathbb{Z}$ classification [5, 6]. The integer invariant $\nu$ counts the number of protected Majorana zero modes on one edge. When interactions are taken into account, the classification collapses to $\mathbb{Z}_8$ [33, 34], i.e. a state with $\nu = 8$, although topologically nontrivial for free fermions, can be trivialized by strong interactions.

Now we consider decorating the fundamental $\mathbb{Z}_4$ domain walls labeled by $\mathbf{g}$ by the $\nu = 2$ 1D FSPT states. Correspondingly, the $\mathbf{g}^2$ domain walls are decorated by the $\nu = 4$ 1D FSPT state, etc. Finally, the $\mathbf{g}^4 = 1$ domain walls are decorated by $\nu = 8$ states which become trivial in the presence of strong interactions, as required by the consistency of the construction. This is also why such a decorated domain wall construction necessarily requires strong interactions to exist.

From the construction, it follows that a defining feature of the edge states in this FSPT phase is that when the $\mathbb{Z}_4$ symmetry is broken, a $\mathbb{Z}_4$ domain wall carries a pair of Majorana zero modes protected by the time-reversal symmetry.

While in principle one can study edge states using the exactly-solvable lattice model, in practice such models are complicated to work with (see for examples Ref. [14] and Ref. [27]).

---

[1] At the domain wall, the time-reversal symmetry is broken so the remaining symmetry is just the fermion parity conservation $\mathbb{Z}_2^f$ and the corresponding symmetry class for free fermions is class D.

In this work we adopt a different approach, ultilizing the connection between SPT phases with internal symmetry and those with crystalline symmetry with the same group structure.

## 2.2 Correspondence with crystalline SPT phases

The one-to-one correspondence between SPT phases with internal and crystalline symmetries was observed in many examples, and recently formalized in Ref. [35]. We provide a heuristic explanation for why this is true, and refer the interested readers to Ref. [35] for a more systematic approach.

Suppose that the low-energy physics of a system of interest can be described by a continuum field theory. It is very common that the continuum field theory enjoys a larger symmetry than the microscopic Hamiltonian, for example discrete lattice translations enhanced to continuous ones. A discrete lattice translation operation is implemented on the fields by the corresponding (actually continuous) one, possibly combined with an internal transformation. However, since the continuous translation itself is a symmetry, the purely internal part of the transformation must be a symmetry of the field theory as well. In other words, one can extract the "internal" action of the lattice translation by simply dropping the coordinate shift. Thus a crystalline symmetry becomes effectively an internal one within the field-theoretical description. We thus expect that the classification of SPT phases with a crystalline symmetry group is the same as those with an internal symmetry as long as the group structures are identical [2].

While the equivalence works at the level of topological classifications, technically it is often the case that crystalline SPT phases are easier to understand thanks to the "block state" construction [36–39]. For example, consider 2D SPT phases protected by $\mathbb{Z} \times G$ where $\mathbb{Z}$ is lattice translation along the $y$ direction and $G$ is an on-site symmetry group. Besides those SPT phases protected by $G$ alone, the rest can all be constructed by stacking 1D states protected by $G$, i.e. there is a 1D SPT state 'per unit length' along $y$.

This argument applies to boundary theories as well. An edge along the same direction preserves the translation (as well as all the internal symmetries). In this construction, the edge is nothing but a chain of end states of the 1D SPT phase which builds up the bulk, and each site transforms projectively under the internal symmetry (i.e. $\mathbb{Z}_2^f \times \mathbb{Z}_2^\mathsf{T}$). We will study an exactly solvable lattice model of this edge, and in particular a critical point described by a (1+1)d Luttinger liquid, which is invariant under continuous translations. One can then derive the symmetry transformations on the low-energy degrees of freedom and extract the "on-site" part of the transformations. We will show that if the 1D building block of the bulk state is chosen to be the $\nu = 2$ 1D FSPT phase, the resulting edge field theory has all the features expected for the edge of an intrinsically interacting FSPT phase with $\mathbb{Z}_4 \times \mathbb{Z}_2^\mathsf{T}$, and the lattice translation is identified with the $\mathbb{Z}_4$ (namely, the "internal" part of the lattice translation has order 4).

Similar methods have been applied to study both bulk and boundary physics of interacting SPT phases in 3D [28, 40, 41].

## 3 The Microscopic Model

We consider a 2D weak topological superconductor, where the bulk is an array of 1D wires in the BDI class. Looking at the edge, we have a 1D chain of Majorana modes:

$$\gamma_i^\dagger = \gamma_i,\ \eta_i^\dagger = \eta_i,\ \gamma_i^2 = \eta_i^2 = 1, i = 1, 2, \ldots, 2N., \tag{1}$$

---

[2] For point-group operations on fermions, additional subtleties occur relating to how the symmetry group is extended by the fermion parity symmetry [28], but this subtlety is not relevant for the kind of symmetry we are interested in.

which satisfy the following algebra:

$$\{\gamma_i, \gamma_j\} = \{\eta_i, \eta_j\} = 2\delta_{ij}, \{\gamma_i, \eta_j\} = 0 \; \forall \; i, j. \tag{2}$$

The time-reversal (TR) symmetry $T_r$ acts as

$$T_r : \begin{pmatrix} \gamma \\ \eta \\ i \end{pmatrix} \longrightarrow \begin{pmatrix} \gamma \\ \eta \\ -i \end{pmatrix}. \tag{3}$$

We can then pairwise combine the $\gamma_i$ and $\eta_i$ into a complex fermion

$$\psi_j = \frac{\gamma_j + i\eta_j}{2}, \tag{4}$$

with canonical commutation relation $\{\psi_i, \psi_j^\dagger\} = \delta_{ij}, \{\psi_i, \psi_j\} = 0$. TR symmetry then becomes an anti-unitary particle-hole transformation:

$$T_r : \psi_j \rightarrow \psi_j^\dagger. \tag{5}$$

Figure 1: Combining $4N$ Majorana edge modes, pairwise, to form $2N$ physical fermions

It is straightforward to check that any Hamiltonian quadratic in $\psi_i, \psi_j^\dagger$ is not allowed as it breaks TR symmetry. This also follows from the $\mathbb{Z}$ classification of non-interacting Hamiltonians in BDI class in 1D. Any Hamiltonian we write down then must be interacting. With interactions, it is known that the classification is reduced to $\mathbb{Z}_8$, i.e. eight Majorana zero modes can be gapped out by quartic interactions without spontaneously breaking the TR-symmetry. For the BDI chain, this gapping mechanism must break translation symmetry as one has to group four sites together. In other words, if one is to find a gapped phase without breaking the TR symmetry, the unit cell must be enlarged at least four times.

To explore the possible phases that can occur on edge, we consider the following TR-invariant Hamiltonian for the boundary chain:

$$H = -\sum_i \left( t\psi_{i+1}^\dagger \psi_{i-1} + \Delta\psi_{i+1}\psi_{i-1} + \text{h.c.} \right)\left( 2\psi_i^\dagger \psi_i - 1 \right). \tag{6}$$

For simplicity we assume both $t$ and $\Delta$ are real in the following. The model possesses translation symmetry; on our physical fermions translation in the transverse direction acts as $T_t : \psi_i \rightarrow \psi_{i+1}$. We will consider closing the chain into a ring with periodic boundary conditions (PBC) (i.e. $\psi_{2N+1} = \psi_1$).

This model is exactly solvable. Employing a Jordan-Wigner (JW) transformation twice we can effectively split the chain in two (even sites and odd sites). One can then think of the model as two copies of a p-wave superconductor, with the caveat that the JW transformation maps a physical fermion to a non-local object in the "free" fermions.

## 3.1 Jordan-Wigner transformation

Recall the JW mapping:

$$\psi_i = \left(\prod_{j=1}^{i-1} \tau_j^z\right) \tau_i^-.$$
(7)

Here $\tau^{x,y,z}$ are Pauli matrices. There is some subtlety involving the the BC conditions of the chains which we will address in a separate section. As an example of the fermion-spin mapping, away from the boundary site one finds

$$\psi_{i+1}^\dagger \psi_{i-1}(2\psi_i^\dagger \psi_i - 1) = \tau_{i+1}^+ \tau_{i-1}^-.$$
(8)

Note that we only have next to nearest neighbor interactions. This will be the case for all the other terms in the Hamiltonian as well. Carrying out the JW mapping on the other terms one arrives at

$$H = -\sum_i \left(t \tau_{i+1}^+ \tau_{i-1}^- + \Delta \tau_{i+1}^- \tau_{i-1}^- + \text{h.c.}\right).$$
(9)

With only next-nearest-neighbor couplings, the Hamiltonian decomposes into two decoupled ones on even and odd sites, respectively. We can further JW transform the two sets (even site and odd site) of spin degrees of freedom resulting in two species of JW fermions. Given the partitioning of the sites into even and odd it makes sense to make this explicit in our notation. Let

$$\tilde{f}_j = \psi_{2j-1}, \quad f_j = \psi_{2j}.$$
(10)

We will refer to $\psi_i$ (as well as $f_j, \tilde{f}_j$) as physical fermions since they are local operators in the original theory. We will similarly define

$$\sigma_j = \tau_{2j}, \tilde{\sigma}_j = \tau_{2j-1}.$$
(11)

For later reference, we give the explicit expressions of the JW fermions $c$ in terms of the physical fermions $f$:

$$c_n = \prod_{j=1}^n (-1)^{\tilde{f}_j^\dagger \tilde{f}_j} f_n, \quad \tilde{c}_n = \prod_{j=1}^{n-1} (-1)^{f_j^\dagger f_j} \tilde{f}_n.$$
(12)

Note that on a given chain our JW fermions do have fermionic statistics but JW fermions from different chains actually commute: $[c_m, \tilde{c}_n] = 0 = [c_m, \tilde{c}_n^\dagger]$.

The Hamiltonian becomes

$$H = \sum_j \left(-t c_{j+1}^\dagger c_j + \Delta c_{j+1} c_j + \text{h.c.}\right) + (c \to \tilde{c}).$$
(13)

## 3.2 Boundary conditions

Define the parity operator

$$P = \prod_{i=1}^{2N} \tau_i^z = \prod_j^N (1 - 2\tilde{f}_j^\dagger \tilde{f}_j)(1 - 2f_j^\dagger f_j).$$
(14)

We can similarly define the parity of the even and odd site chains

$$P_1 = \prod_{j=1}^N (1 - 2\tilde{f}_j^\dagger \tilde{f}_j), \quad P_2 = \prod_{j=1}^N (1 - 2f_j^\dagger f_j).$$
(15)

Note $P$, $P_1$ and $P_2$ all commute with $H$. Let $\mu_f$ and $\mu_b$ denote the boundary conditions on the physical fermions and the JW spin degrees of freedom, respectively. Recall that we are assuming a PBC in the fermionic variables so $\mu_f = 1$. Consider one of the boundary terms of our original $H$:

$$
\begin{aligned}
\tilde{f}_{N+1}^\dagger \tilde{f}_N = \mu_f \tilde{f}_1^\dagger \tilde{f}_N &= \mu_f \tau_1^+ \prod_{i=1}^{2N-2} \tau_i^z \tau_{2N-1}^- \\
&= \tau_{2N+1}^+ P \prod_{i=1}^{2N-2} \tau_i^z \tau_{2N-1}^- \\
&= -\mu_b P \tau_1^+ \prod_{i=1}^{2N-2} \tau_i^z \tau_{2N-1}^- \,.
\end{aligned}
\tag{16}
$$

Therefore we have

$$
\mu_f = -\mu_b P \,.
\tag{17}
$$

When we split the spin chain in two(even sites and odd sites), the resultant chains clearly inherit the same BC, that is $\mu_{b_1} = \mu_b = \mu_{b_2}$, where $\mu_{b_i}$ is the BC of the spin degrees of freedom on chain $i$.

Denote the BCs for the JW operators $\tilde{c}$ and $c$ by $\mu_{f_1}$ and $\mu_{f_2}$. Then a similar argument shows

$$
\mu_{f_i} = -P_i \mu_{b_i} = -P_i \mu_b = P_i P \mu_f \,,
\tag{18}
$$

so $\mu_{f_1} = P_2 \mu_f$ and $\mu_{f_2} = P_1 \mu_f$. We have imposed a PBC on the physical fermions $f$ ($\mu_f = 1$) thus:

$$
\mu_{f_1} = P_2 \text{ and } \mu_{f_2} = P_1 \,.
\tag{19}
$$

Note we can divide up our Hilbert space into four parity sectors $(P_1, P_2) = (\pm 1, \pm 1)$ or $(\pm 1, \mp 1)$.

# 4 Phase Diagram: Ising analysis

Since the model is supposed to describe an anomalous edge, the ground state can not be non-degenerate. Due to the one dimensional nature it is either gapless, or gapped with spontaneous breaking of the symmetries. In this section we analyze the gapped phases of the model Eq. (6). After the JW transformation, the Hamiltonian decomposes into two Kitaev chains, which are gapped as long as both $t$ and $\Delta$ are nonzero. We thus expect that the symmetries must be spontaneously broken. To work out the symmetry breaking properties and gain some intuition about the edge theory, we come back to the spin representation and consider the Ising point: $|t| = |\Delta|$. The behavior at the Ising point should apply to other values of $\Delta$ with the same sign since the gap remains open.

Our spin Hamiltonian is

$$
H = -\sum_i \left[ (\Delta + t) \tau_{i-1}^x \tau_{i+1}^x + (t - \Delta) \tau_{i-1}^y \tau_{i+1}^y \right].
\tag{20}
$$

We know from the properties of the JW transform on a closed chain that $\mu_b = -P$, this will emerge as the natural choice from the energetics of the ground state as well.

Symmetry transformations of the spin variables can be easily derived:

$$
\begin{aligned}
T_r : \tau_i^\pm &\to (-1)^{i-1} \tau_i^\mp, \quad \tau_i^z \to -\tau_i^z, \\
T_t : \tau_i &\to \tau_{i+1} \,.
\end{aligned}
\tag{21}
$$

To diagnose the symmetry breaking, we will work with the order parameter $\tilde{f}_i^\dagger f_i = \tilde{\sigma}_i^+ \tilde{\sigma}_i^z \sigma_i^-$. For a given $t$ we may consider the two Ising points: $\Delta = t$, which corresponds to $H = -2t \sum_i (\sigma_i^x \sigma_{i+1}^x + \tilde{\sigma}_i^x \tilde{\sigma}_{i+1}^x)$, and $\Delta = -t$ which corresponds to $H = -2t \sum_i (\sigma_i^y \sigma_{i+1}^y + \tilde{\sigma}_i^y \tilde{\sigma}_{i+1}^y)$. With regards to the order parameter, we are really working with its projection onto the ground state:

$$\tilde{f}_i^\dagger f_i = \tilde{\sigma}_i^+ \tilde{\sigma}_i^z \sigma_i^- \cong \begin{cases} \tilde{\sigma}_i^x \sigma_i^x, & \Delta = t \\ \tilde{\sigma}_i^y \sigma_i^y, & \Delta = -t \end{cases}, \tag{22}$$

where we take $\cong$ to mean equal at the level of projecting onto the ground state space. Consider the transformation properties of the order parameter (and its ground state projection) under $T_t$ and $T_r$:

$$T_t : \tilde{f}_i^\dagger f_i \to f_i^\dagger \tilde{f}_{i+1} \cong \begin{cases} \sigma_i^x \tilde{\sigma}_{i+1}^x, & \Delta = t \\ \sigma_i^y \tilde{\sigma}_{i+1}^y, & \Delta = -t \end{cases} \tag{23}$$

and

$$T_r : \tilde{f}_i^\dagger f_i \to -f_i^\dagger \tilde{f}_i \cong \begin{cases} -\tilde{\sigma}_i^x \sigma_i^x, & \Delta = t \\ -\tilde{\sigma}_i^y \sigma_i^y, & \Delta = -t \end{cases}. \tag{24}$$

Below we work out the symmetry breaking properties for the $\Delta = t$ case. From this analysis, it is clear that the symmetry breaking properties of the $\Delta = -t$ case are the same. Flipping the sign of $\Delta$ simply rotates between $\sigma^x$ and $\sigma^y$ in the Hamiltonian and in the projection of the order parameter on the ground state space. The upshot of this is that that the symmetry breaking of the ground state only depends on the sign of $t$.

On general grounds, we expect that $T_r$ will always be broken, as an Ising-like Hamiltonian can at most induce translation symmetry breaking with a doubled unit cell. Whether $T_t$ is broken depends on the sign of the coupling $t$, i.e. ferromagnetic or anti-ferromagnetic. Below we determine the ground state(s) for the different cases of sign of $t$ and even/oddness of $N$.

## 4.1 $N$ even

Let $t = \Delta$. One can then basically read off the ground states. In each case the spin chains will have PBC; $\mu_B = 1$ means $P = -1$.

### 4.1.1 $t < 0$

$t < 0$ means we are in a staggered anti-ferromagnetic phase; site $i$ will anti-align with site $i + 2$. Thus we expect that the translation symmetry is broken spontaneously.

Our ground state space will be constructed from the states

$$\{|\uparrow\uparrow\downarrow\downarrow \dots \downarrow\downarrow\rangle, |\uparrow\downarrow\downarrow\uparrow \dots \downarrow\uparrow\rangle, |\downarrow\uparrow\uparrow\downarrow \dots \uparrow\downarrow\rangle, |\downarrow\downarrow\uparrow\uparrow \dots \uparrow\uparrow\rangle\}. \tag{25}$$

Here $|\uparrow\rangle/|\downarrow\rangle$ is the eigenstate of $\tau^x$ with eigenvalue $+/-$. The BC-parity relationship can be used to quickly read off the ground state. Note that $P = \prod_i \sigma_i^z$ which, in the $x$-basis just flips the spin at every site. Since parity is a good quantum number, we have a $d = 2$ ground state space: with basis

$$\begin{aligned} |+\rangle &= |\uparrow\uparrow\downarrow\downarrow \dots\rangle - |\downarrow\downarrow\uparrow\uparrow \dots\rangle \\ |-\rangle &= |\uparrow\downarrow\downarrow\uparrow \dots\rangle - |\downarrow\uparrow\uparrow\downarrow \dots\rangle, \end{aligned} \tag{26}$$

with parity eigenvalue $-1$. Now lets compute expectation values of our order parameter. In the ground states, we find

$$\langle \pm | \tilde{\sigma}_i^x \sigma_i^x | \pm \rangle = \pm 1 = -\langle \pm | \sigma_i^x \tilde{\sigma}_{i+1}^x | \pm \rangle. \tag{27}$$

Since $\tilde{\sigma}_i^x \sigma_i^x$ is odd under $T_r$, the TR symmetry is spontaneously broken. From Eq. (27) it is also clear that $T_t$ is broken. Therefore, both $T_t$ and $T_r$ are broken, while their product, $T_t T_r$, is preserved.

### 4.1.2 $t > 0$

Here we are in a staggered ferromagnetic phase. A basis for our ground state, which must have $P = -1$, is

$$
\begin{aligned}
|+\rangle &= |\uparrow\uparrow\uparrow\uparrow ....\rangle - |\downarrow\downarrow\downarrow\downarrow .....\rangle \\
|-\rangle &= |\uparrow\downarrow\uparrow\downarrow ....\rangle - |\downarrow\uparrow\downarrow\uparrow .....\rangle
\end{aligned}
\tag{28}
$$

and we see

$$
\langle\pm|\,\tilde{\sigma}_i^x \sigma_i^x\,|\pm\rangle = \pm 1 = \langle\pm|\,\sigma_i^x \tilde{\sigma}_{i+1}^x\,|\pm\rangle\,,
\tag{29}
$$

suggesting $T_t$ is not broken, as one expects.

## 4.2 $N$ odd

### 4.2.1 $t < 0$

Again we are in a staggered anti-ferromagnetic phase but the oddness of the split chains requires an APBC: Our ground state space will be constructed from the states

$$
\{|\uparrow\uparrow\downarrow\downarrow ... \uparrow\uparrow\rangle, |\uparrow\downarrow\downarrow\uparrow ... \uparrow\downarrow\rangle, |\downarrow\uparrow\uparrow\downarrow ... \downarrow\uparrow\rangle, |\downarrow\downarrow\uparrow\uparrow ... \downarrow\downarrow\rangle\}
\tag{30}
$$

but now $P = 1$. The $P = 1$ ground state basis is

$$
\begin{aligned}
|+\rangle &= |\uparrow\uparrow\downarrow\downarrow ...\rangle + |\downarrow\downarrow\uparrow\uparrow ...\rangle\,, \\
|-\rangle &= |\uparrow\downarrow\downarrow\uparrow ...\rangle + |\downarrow\uparrow\uparrow\downarrow ...\rangle\,.
\end{aligned}
\tag{31}
$$

Checking the order parameter expectation values we see:

$$
\begin{aligned}
\langle\pm|\,\tilde{\sigma}_i^x \sigma_i^x\,|\pm\rangle &= \pm 1 = -\langle\pm|\,\sigma_i^x \tilde{\sigma}_{i+1}^x\,|\pm\rangle\,, \\
\langle\pm|\,T_r \tilde{\sigma}_i^x \sigma_i^x T_r^{-1}\,|\pm\rangle &= -\langle\pm|\,\tilde{\sigma}_i^x \sigma_i^x\,|\pm\rangle\,.
\end{aligned}
\tag{32}
$$

As in the $N$ even case both $T_t$ and $T_r$ are broken, while their product $T_t T_r$ is not.

### 4.2.2 $t > 0$

This case turns out to be the same as the $N$ even one: only $T_r$ is broken, because of the ferromagnetic coupling.

# 5 Low-energy Field Theory

The model becomes gapless at $\Delta = 0$:

$$
H_0 = -t \sum_j (c_{j+1}^\dagger c_j + \text{h.c.}) + (c \to \tilde{c})\,.
\tag{33}
$$

We will assume $t > 0$. At this point, the Hamiltonian is simply free JW-fermions hopping on the chains and no symmetries are broken. The TR symmetry fixes the chemical potential at 0, i.e. half-filling; so $k_F = \frac{\pi}{2}$. Further interactions can be incorporated by bosonization. However, one must keep in mind that $c$ and $\tilde{c}$ are highly non-local in terms of physical fermions. In the following we will work out the bosonized theory for this gapless point. Of particular importance is how the low-energy fields transform under the global symmetries, and how physical fermions are represented in the low-energy theory.

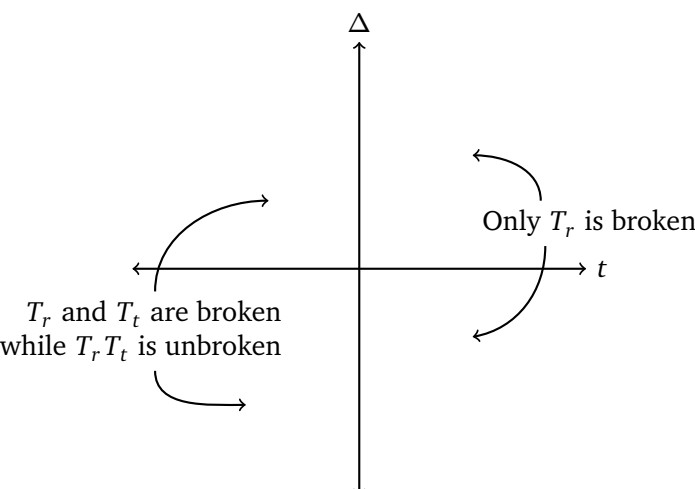

Figure 2: Phase diagram of the model Hamiltonian Eq. (6). The symmetry breaking pattern only depends on the sign of $t$.

## 5.1 Bosonization

Following the standard bosonization prescription, we linearize the spectrum around the two Fermi points $\pm\frac{\pi}{2}$, and define chiral fields:

$$c_{k_F+k} = c_{R,k}, c_{-k_F+k} = c_{L,k}, \tag{34}$$

where $R/L$ stand for right/left moving. Introduce a continuum field $\psi(x) \sim c_x$, we can write

$$\psi(x) = e^{i\frac{\pi}{2}x}\psi_R(x) + e^{-i\frac{\pi}{2}x}\psi_L(x), \tag{35}$$

where the chiral fields are defined as

$$\psi_{R/L}(x) \sim \frac{1}{\sqrt{N}} \sum_k e^{ikx} c_{R/L,k}. \tag{36}$$

There is a similar field $\tilde{\psi}$ on the other chain. In the large size limit we see $\{\psi(x), \psi^\dagger(y)\} = 2\pi\delta(x-y) = \{\tilde{\psi}(x), \tilde{\psi}^\dagger(y)\}$ while fields from different chains commute i.e $[\psi(x), \tilde{\psi}(y)] = [\psi(x), \tilde{\psi}^\dagger(y)] = 0$.

Now we can bosonize the fields [42]:

$$\psi_{L/R}(x) \sim e^{i[\theta(x)\pm\phi(x)]}, \tag{37}$$

where the bosonic fields satisfy the canonical commutation relation $[\phi(x), \partial_y\theta(y)] = i\pi\delta(x-y)$. $\tilde{\theta}, \tilde{\phi}$ are similarly defined. Note that in our definition $\phi, \theta$ and $\tilde{\phi}, \tilde{\theta}$ commute, reflecting the fact that our JW fermions from different chains commute. Anti-commutation between $\psi$ and $\tilde{\psi}$ can be re-enforced by introducing Klein factors, but they are not necessary for our purpose. The non-interacting Hamiltonian can be expressed in terms of the bosonic fields $\Phi = (\phi, \theta, \tilde{\phi}, \tilde{\theta})^T$:

$$H = \frac{v}{2\pi}\int dx \, [(\partial_x\phi)^2 + (\partial_x\theta)^2] + \frac{v}{2\pi}\int dx \, [(\partial_x\tilde{\phi})^2 + (\partial_x\tilde{\theta})^2], \tag{38}$$

where $v = ta_0$. The theory is a $c = 2$ Luttinger liquid. The Luttinger parameter is 1 in the free theory, and can be tuned to other values when density-density interactions are included.

While the bosonization is fairly straightforward, an important ingredient of the low-energy theory is how physical electrons are represented, which determine the allowed operator content. In terms of the bosonic fields, physical fermions are given by attaching the JW string to $\psi(x)$ and $\tilde{\psi}(x)$:

$$e^{\pm i\tilde{\phi}\pm\phi\pm\theta}, e^{\pm i\phi\pm\tilde{\phi}\pm\tilde{\theta}}. \tag{39}$$

Their combinations give all physical operators. This is a nontrivial requirement, forbiding operators like $\psi^\dagger\tilde{\psi}$. One may understand the constraints as a gauge symmetry, which has important consequences for boundary conditions. One can show that a general vertex operator $e^{i\mathbf{l}^T\Phi}$ is physical if and only if both $l_1 + l_2 + l_4$ and $l_3 + l_2 + l_4$ are even integers. Furthermore, if $e^{i\mathbf{l}^T\Phi}$ is a bosonic operator, then $l_1 + l_2 + l_3 + l_4$ must be even, so $l_1, l_3$ and $l_2 + l_4$ are all even.

## 5.2 Symmetry transformations of $\Phi$

What distinguishes the field theory from an ordinary 1D quantum wire is their anomalous transformation properties under the symmetries. The lattice model has translation whose generator we denote by $t$, and time-reversal symmetry generated by $r$. Notice that $r^2 = \mathbb{1}$ and $rt = tr$. In addition, the model also has U(1) charge conservation, but it is not relevant.

From the lattice model (See Appendix C for derivation)

$$T_r : \begin{pmatrix} c_{k,L/R} \\ \tilde{c}_{k,L/R} \end{pmatrix} \rightarrow \begin{pmatrix} c^\dagger_{k,R/L} \\ -\tilde{c}^\dagger_{k,R/L} \end{pmatrix}, \tag{40}$$

so our fields transform as

$$T_r \begin{pmatrix} \psi_{L/R} \\ \tilde{\psi}_{L/R} \end{pmatrix} \rightarrow \begin{pmatrix} \psi^\dagger_{R/L} \\ -\tilde{\psi}^\dagger_{R/L} \end{pmatrix} \Rightarrow T_r : \begin{pmatrix} \phi \\ \theta \\ \tilde{\phi} \\ \tilde{\theta} \end{pmatrix} \rightarrow \begin{pmatrix} -\phi \\ \theta \\ -\tilde{\phi} \\ \tilde{\theta} + \pi \end{pmatrix}. \tag{41}$$

Our bosonization procedure (definitions of L/R moving fields etc) has assumed $t > 0$ but one can study the $t < 0$ using the same conventions as Sec. 5.1 by mapping $t \rightarrow -t$ via the unitary transformation $(c_i, \tilde{c}_i) \rightarrow ((-1)^i c_i, (-1)^i \tilde{c}_i)$. Note that boundary terms transform like $c^\dagger_N c_1 \rightarrow (-1)^{N-1} c^\dagger_N c_1$. For $N$ odd, we see that the boundary condition is flipped in addition to the sign of $t$.

With this in mind we can work out the translation transformation properties of $\Phi$ given $T_t : \tilde{f}_i \rightarrow f_i$ etc. Recall that

$$c_i \sim e^{-i\frac{\pi}{2}x}e^{i(\theta+\phi)} + e^{i\frac{\pi}{2}x}e^{i(\theta-\phi)},$$
$$\tilde{c}_i \sim e^{-i\frac{\pi}{2}x}e^{i(\tilde{\theta}+\tilde{\phi})} + e^{i\frac{\pi}{2}x}e^{i(\tilde{\theta}-\tilde{\phi})}. \tag{42}$$

For $t > 0$, $c_i \rightarrow \tilde{c}_{i+1}$ and $\tilde{c}_i \rightarrow c_i$ under $T_t$ gives

$$T_t : \begin{pmatrix} \phi \\ \theta \\ \tilde{\phi} \\ \tilde{\theta} \end{pmatrix} \rightarrow \begin{pmatrix} \tilde{\phi} - \frac{\pi}{2} \\ \tilde{\theta} \\ \phi \\ \theta \end{pmatrix}. \tag{43}$$

For $t < 0$, $c_i \rightarrow -\tilde{c}_{i+1}$ and $\tilde{c}_i \rightarrow c_i$ giving

$$T_t : \begin{pmatrix} \phi \\ \theta \\ \tilde{\phi} \\ \tilde{\theta} \end{pmatrix} \rightarrow \begin{pmatrix} \tilde{\phi} - \frac{\pi}{2} \\ \tilde{\theta} + \pi \\ \phi \\ \theta \end{pmatrix}. \tag{44}$$

We have suppressed the coordinate change associated with the translation.

Notice that in all cases we have $T_r^2$ and $T_t^4$ acting as the identity on bosonic fields. However, $T_r$ and $T_t$ do not commute when acting on $\Phi$, which seems to contradict the fact that the symmetry group is $\mathbb{Z}_2^{\mathsf{T}} \times \mathbb{Z}_4$. The reason for the inconsistency is because $\phi, \theta, \tilde{\phi}$ and $\tilde{\theta}$ are not local fields. As we will see below, when acting on local degrees of freedom $T_t$ and $T_r$ do represent the group faithfully.

## 5.3 K matrix formulation

We have derived a low-energy theory from the lattice model. Here we discuss an alternative formulation using K matrix [43–48], which has the advantage that only physical degrees of freedom (allowing chiral ones) appear. First we give a brief overview of K matrix theory. A general (chiral or non-chiral) Luttinger liquid is described by the following Lagrangian:

$$\mathcal{L} = \frac{1}{4\pi} \sum_{IJ} K_{IJ} \partial_t \phi_I \partial_x \phi_J - \frac{1}{4\pi} \sum_{IJ} V_{IJ} \partial_x \phi_I \partial_x \phi_J - \cdots \tag{45}$$

Here $K$ is a symmetric integer matrix, which determines the commutation relations between fields: $[\phi_I(y), \partial_x \phi_J(x)] = 2\pi i (K^{-1})_{IJ} \delta(y-x)$. Since we are considering an edge of a short-range entangled bulk without fractionalized excitations, we require $\det K = \pm 1$. For such unimodular K matrices, all excitations $e^{i\phi_i}$ are physical. The non-universal $V$ matrix determines velocities of bosonic modes as well as scaling dimensions of operators.

A general symmetry transformation $T_g$ takes the following form

$$T_g^{-1} \phi_I T_g = \sum_j (W_g)_{IJ} \phi_J + (\delta \phi_g)_I. \tag{46}$$

To preserve the commutation relations the integer matrix $W_g$ must satisfy

$$W_g K^{-1} W_g^T = \pm K^{-1}, \tag{47}$$

$+/-$ for unitary/anti-unitary transformations. In addition, they must obey group multiplication laws: $W_g W_h = W_{gh}$,

The K matrix for a Luttinger liquid is generally not uniquely defined because one can make a change of variable: $\phi_I = \sum_J W_{IJ} \phi'_J$, where $W$ is an invertible integer matrix (i.e. $|\det W| = 1$). For the new fields, the K matrix becomes $\tilde{K}' = W^T K W$, and

$$W'_g = W^{-1} W_g W, \delta \phi'_g = W^{-1} \delta \phi_g. \tag{48}$$

To obtain such a description, we first find a basis for local operators in the theory. They can be chosen as $\phi_I = \mathbf{1}_I^T \Phi$ with

$$\begin{aligned}
\mathbf{l}_1 &= (1, 1, 1, 0), \\
\mathbf{l}_2 &= (1, 0, 1, 1), \\
\mathbf{l}_3 &= (-1, 1, 1, 0), \\
\mathbf{l}_4 &= (1, 0, -1, 1).
\end{aligned} \tag{49}$$

Their commutation relations are given by the following K matrix:

$$K = \begin{pmatrix} 1 & -1 & 0 & 1 \\ -1 & 1 & 1 & 0 \\ 0 & 1 & -1 & -1 \\ 1 & 0 & -1 & -1 \end{pmatrix}. \tag{50}$$

Symmetry properties can be readily obtained from Eq. (41) and (43). We find that under TR symmetry

$$W_r = \begin{pmatrix} 0 & -1 & 1 & 1 \\ -1 & 0 & 1 & 1 \\ 1 & -1 & 0 & 1 \\ -1 & 1 & 1 & 0 \end{pmatrix}, \delta\phi_r = \begin{pmatrix} 0 \\ \pi \\ 0 \\ \pi \end{pmatrix}, \tag{51}$$

and under lattice translation:

$$W_t = \begin{pmatrix} 0 & 1 & 0 & 0 \\ 1 & 0 & 0 & 0 \\ 0 & 0 & 0 & 1 \\ 0 & 0 & 1 & 0 \end{pmatrix}, \delta\phi_t = -\frac{\pi}{2} \begin{pmatrix} 1 \\ 1 \\ -1 \\ 1 \end{pmatrix}. \tag{52}$$

We can further simplify the K matrix. A change of variables $\phi = W\phi'$ with

$$W = \begin{pmatrix} 1 & 0 & 0 & 0 \\ 1 & 0 & 1 & -1 \\ 0 & 0 & 0 & 1 \\ 1 & 1 & 0 & -1 \end{pmatrix}, \tag{53}$$

brings $K$ into the standard diagonal form:

$$W^T K W = \begin{pmatrix} \sigma_z & 0 \\ 0 & \sigma_z \end{pmatrix}. \tag{54}$$

This is expected from the general classification of non-chiral, unimodular K matrices. Using Eq. (48) we obtain

$$W_r' = \begin{pmatrix} 0 & 1 & -1 & 1 \\ 1 & 0 & 1 & -1 \\ 1 & 1 & 0 & -1 \\ 1 & 1 & -1 & 0 \end{pmatrix}, \delta\phi_r' = \begin{pmatrix} 0 \\ \pi \\ \pi \\ 0 \end{pmatrix}, \tag{55}$$

and

$$W_t' = \begin{pmatrix} 1 & 0 & 1 & -1 \\ 0 & 1 & -1 & 1 \\ 1 & 1 & -1 & 0 \\ 1 & 1 & 0 & -1 \end{pmatrix}, \delta\phi_t' = \frac{\pi}{2} \begin{pmatrix} -1 \\ 1 \\ 1 \\ 1 \end{pmatrix}. \tag{56}$$

Using the matrix representations, one can check that $T_r^2 = T_t^4 = \mathbb{1}, T_r T_t = T_t T_r$ and $T_t^2 \neq \mathbb{1}, P$ where $P$ is the global fermion parity, which shows that the symmetry group is indeed $\mathbb{Z}_2^T \times \mathbb{Z}_4$.

While the K matrix now is the same as the one for free fermions, we emphasize that it does not mean the theory is free after the basis transformation, because the symmetry transformations become complicated. For a free theory, we expect that a $n$-body operator remains $n$-body under symmetry transformations, which is not the case for $W_r'$ and $W_t'$: for example, they map a 1-body operator to a 3-body one. One can further check that no other basis transformations can bring $W_t$ and $W_r$ into a form expected for a free theory, while keeping $K$ the same.

It is crucial that the K matrix is $4 \times 4$, which allows non-trivial transformations such as $W_r'$ and $W_t'$. We show in the appendix that $2 \times 2$ K matrix can not describe such an edge. In fact, we prove that within the K matrix framework, there are no nontrivial fermionic SPT phases with $2 \times 2$ K matrix. Therefore the theory found here is in some sense "minimal".

Although we have provided a completely local description of the effective theory, in the following we will still work with the formulation given in Sec. 5.2, as it is easier to relate to the lattice model.

## 5.4 Gapped phases in the bosonic field theory

With a complete low-energy gapless theory, we can explore effects of more complicated interactions to understand its stability. Here we first consider the stability with respect to gapping perturbations of null-vector type [44]. For U(1) bosons, generic local interactions are given by vertex operators of the form $e^{i\mathbf{l}^T\Phi}$, where $\mathbf{l}$ is an integer vector. Given that we are working with non-local variables, additional constraints must be placed on $\mathbf{l}$ to ensure locality, as discussed in Sec. 5.1.

In an effort to gap out the theory we can consider adding Higgs terms of the form [25, 44–46, 49] $\sum_a U_a \cos(\mathbf{l}_a^T\Phi - \alpha_a)$ with $\mathbf{l}_a \in \mathbb{Z}^4$. Restricting our attention to the gapping terms which respect time reversal and translation symmetry provides a verification of the robustness of the gapless edge and the nontrivial symmetry-protected topological order of the bulk. To gap out the edge modes, it is sufficient to choose $\{\mathbf{l}_a\}$ as a set of linearly independent null vectors, namely they satisfy

$$[\mathbf{l}_a^T\Phi, \mathbf{l}_b^T\Phi] = 0 \tag{57}$$

for all $a, b$. Then in the limit of large $U_a$, all $\mathbf{l}_a^T\Phi$ simultaneously acquire finite expectation values to minimize the cosine potentials. Since there are two conjugate pairs of bosonic fields, two null vectors are needed to freeze all degrees of freedom.

Our basic tactic is the following: consider a set of symmetry-preserving, independent gapping terms $\{\cos(\mathbf{l}_a^T\Phi - \alpha_a)\}$ for a set of null vectors $\mathbf{l}_a$. We then check whether there exists any local, elementary field $\mathbf{v}^T\Phi$ that acquires a finite expectation value in the ground state (meaning that a certain linear combination of $\mathbf{l}_a$'s is a multiple of $\mathbf{v}$). If these fields transform non-trivially under the symmetry transformations, then the ground state spontaneously breaks the symmetry. A more systematic treatment can be found in Ref. [50].

### 5.4.1 Continuum limit of the solvable model

Before considering general gapping terms, let us analyze the continuum limit of the pairing term in the lattice model [51]:

$$\sum_j \Delta c_{j+1} c_j + \text{h.c.} \sim \Delta \int_0^L dx \left[ \psi(x+a)\psi(x) + \text{h.c} \right]$$
$$= \Delta \int_0^L dx \left[ e^{-i\frac{\pi}{2}} e^{i2\theta(x)} + \text{h.c.} \right]. \tag{58}$$

Here $a$ is the short-distance cutoff.

So the superconducting term $-\Delta(c_{j+1}c_j + \tilde{c}_{j+1}\tilde{c}_j) + \text{h.c.}$ becomes $\Delta(\sin 2\theta + \sin 2\tilde{\theta})$. Without loss of generality, assume $\Delta > 0$. In the large $L$ limit, $\theta$ is pinned at the minima of $\Delta \sin 2\theta$, namely $\theta = -\frac{\pi}{4}$ or $\frac{3\pi}{4}$. Recall though that the physical ground states should have definite total fermion parity. One can check that $P_1 = e^{i\int_0^L \partial_x \tilde{\phi}}$ and $P_2 = e^{i\int_0^L \partial_x \phi}$, thus:

$$P = \exp\left( i \int_0^L \partial_x \tilde{\phi} + \partial_x \phi \right). \tag{59}$$

From the bosonic commutation relations we see

$$P\theta P^{-1} = \theta + \pi, \qquad P\tilde{\theta}P^{-1} = \tilde{\theta} + \pi. \tag{60}$$

The Hamiltonian conserves both $P_1$ and $P_2$.

We know the parity of our ground state from the lattice model but it is useful to derive it from the field theory. The physical fermion $e^{i(\tilde{\phi}+\phi+\theta)}$ satisfies PBC, which means

$$
\begin{aligned}
e^{i[\tilde{\phi}(L)+\phi(L)+\theta(L)]} &= e^{i[\int_0^L \partial_x(\tilde{\phi}+\phi+\theta)+(\tilde{\phi}(0)+\phi(0)+\theta(0))]} \\
&= -e^{i\int_0^L \partial_x(\tilde{\phi}+\phi+\theta)} e^{i[\tilde{\phi}(0)+\phi(0)+\theta(0)]}.
\end{aligned}
\tag{61}
$$

Because in the ground state manifold $\theta$ is pinned, we see that the BC is $-P_1 P_2$. Thus we find $P = P_1 P_2 = -1$. (For $t > 0$ and odd $N$, it is the opposite). Similarly we find

$$
\begin{aligned}
\psi_{L/R}(L) &= -e^{i(\int_0^L dx \partial_x \theta \pm \int_0^L dx \partial_x \phi)}\psi_{L/R}(0) \\
&= -P_2 \psi_{L/R}(0),
\end{aligned}
\tag{62}
$$

in accordance with the lattice result $c_{N+1} = P_1 c_1$.

Now we work out the ground states for the field theory and check the symmetry breaking pattern. For the sake of explicitness consider chain 2. We can form the parity (i.e. $P_2$) eigenstates $|\pm\rangle_2 = \left|\frac{-\pi}{4}\right\rangle_2 \pm \left|\frac{3\pi}{4}\right\rangle_2$, where $P_2 |\pm\rangle_2 = \pm |\pm\rangle_2$. The analysis of chain 1 is identical. The ground state space of the full chain is spanned by $|\pm\rangle_1 |\pm\rangle_2$, subject to the constraint of a fixed total fermion parity. For $t > 0$, we have shown that $P = -1$, so the two states are $|+\rangle_1 |-\rangle_2$ and $|-\rangle_1 |+\rangle_2$. It is convenient to form the following superpositions:

$$
|\pm\rangle = |+\rangle_1 |-\rangle_2 \pm |-\rangle_1 |+\rangle_2.
\tag{63}
$$

In terms of $\theta, \tilde{\theta}$ eigenstates:

$$
\begin{aligned}
|+\rangle &= \left|\frac{-\pi}{4}\right\rangle_1 \left|\frac{-\pi}{4}\right\rangle_2 - \left|\frac{3\pi}{4}\right\rangle_1 \left|\frac{3\pi}{4}\right\rangle_2 \\
|-\rangle &= \left|\frac{3\pi}{4}\right\rangle_1 \left|\frac{-\pi}{4}\right\rangle_2 - \left|\frac{-\pi}{4}\right\rangle_1 \left|\frac{3\pi}{4}\right\rangle_2.
\end{aligned}
\tag{64}
$$

Now

$$
T_r : (\theta, \tilde{\theta}) \to (\theta, \tilde{\theta} + \pi)
$$

meaning $T_r : |\pm\rangle \to -|\mp\rangle$ suggesting $T_r$ is broken.

The symmetry breaking can also be detected by an order parameter. In this case, the order parameter is just $\cos(\theta - \tilde{\theta})$, which is odd under $T_r$ but invariant under $T_t$. Its expectation value on $|\pm\rangle$ is $\pm 1$. On the other hand, $\sin(\theta - \tilde{\theta})$ is also odd under translation but its expectation value vanishes.

In the lattice theory $T_t$ breaking depended on the sign of $t$ so we should expect the same behavior in the field theory. Recall

$$
T_t : (\theta, \tilde{\theta}) \to \begin{cases} (\tilde{\theta}, \theta) & t > 0 \\ (\tilde{\theta} + \pi, \theta) & t < 0 \end{cases}.
\tag{65}
$$

We can see that the field theory reproduces the symmetry breaking properties of the lattice. The same result is seen in the odd case with the small adjustment that in the $t > 0$ case our $P = 1$ ground states are given by $|\pm\rangle = |+\rangle_1 |+\rangle_2 \pm |-\rangle_1 |-\rangle_2$.

### 5.4.2 General gapping terms

With the above special case worked out we can now consider general gapping terms. We will focus on the $t > 0$ phase and results for the $t < 0$ phase are very similar. Recall how $\Phi$

transforms under $T_r$ and $T_t$:

$$T_t : \begin{pmatrix} \phi \\ \theta \\ \tilde{\phi} \\ \tilde{\theta} \end{pmatrix} \rightarrow \begin{pmatrix} \tilde{\phi} + \frac{\pi}{2} \\ \tilde{\theta} \\ \phi \\ \theta \end{pmatrix} \quad \text{and} \quad T_r : \begin{pmatrix} \phi \\ \theta \\ \tilde{\phi} \\ \tilde{\theta} \end{pmatrix} \rightarrow \begin{pmatrix} -\phi \\ \theta \\ -\tilde{\phi} \\ \tilde{\theta} + \pi \end{pmatrix}. \tag{66}$$

As discussed already, if our goal is to investigate the gapability of the model we need to consider something like $\delta \mathcal{L} = U_1 \cos(\mathbf{1}_1^T \Phi - \alpha_1) + U_2 \cos(\mathbf{1}_2^T \Phi - \alpha_2)$. Verifying that any symmetry allowed gapping term introduces spontaneous breaking or gapless modes amounts to working through all the allowed cases. We give a proof of the all the cases in Appendix D. Here we will show a few examples to demonstrate the approach.

Let us first consider the case in which each gapping term transforms trivially under both of the symmetries:

$$T_g^{-1} \cos(\mathbf{1}^T \Phi - \alpha) T_g = \cos(\mathbf{1}^T \Phi - \alpha), g = t/r, \tag{67}$$

let $\mathbf{1}^T = (a, b, c, d)$, then acting with symmetry operators on $\cos(a\phi + b\theta + c\tilde{\phi} + d\tilde{\theta} - \alpha)$ one can derive constraints on the vector $\mathbf{1}$. It follows, via $T_t$ symmetry, that $a = \pm c$ for example. We summarize these constraints in the following table:

| Symmetry | Vector constraint | Phase constraint |
|----------|-------------------|------------------|
| $T_t$ | $a = \pm c$ and $b = \pm d$ | $a \in 4\mathbb{Z}$ |
| $T_r$ | $a, c = 0$ or $b, d = 0$ | $d \in 2\mathbb{Z}$ |

From the table one has gapping terms of the form $\cos(4n(\phi \pm \tilde{\phi}))$ or $\cos(2m(\theta \pm \tilde{\theta}))$ which condense. Some fraction of these correspond to physical operators which break $T_t$ and $T_r$ respectively. For example; for $\cos 4n(\phi \pm \tilde{\phi})$, the order parameter $\cos 2(\phi \pm \tilde{\phi})$ has a finite expectation value and breaks translation symmetry.

Now consider the situation in which $T_t$ exchanges the gapping terms and $T_r$ does not. We have an interaction of the form

$$U_1 [\cos(a\phi + b\theta + c\tilde{\phi} + d\tilde{\theta} - \alpha) + \cos(c\phi + d\theta + a\tilde{\phi} + b\tilde{\theta} + \frac{a\pi}{2} - \alpha)]. \tag{68}$$

There are only two Higgs terms so acting with $T_t$ twice must generate a phase of $2n\pi$.

| Symmetry | Vector constraint | Phase constraint |
|----------|-------------------|------------------|
| $T_t$ | | $a + c \in 4\mathbb{Z}$ |
| $T_r$ | $a, c = 0$ or $b, d = 0$ | $b, d \in 2\mathbb{Z}$ |

If $a, c = 0$, because both $b$ and $d$ are even we write $b = 2m, d = 2n$. Then $\delta L \sim \cos(2(m\theta + n\tilde{\theta}) - \alpha) + \cos(2(n\theta + m\tilde{\theta}) - \alpha)$. For $m = \pm n$ these two terms collapse into a single one, meaning the edge has a gapless mode. Otherwise, we can combine the two arguments to get $2(m + n)(\theta + \tilde{\theta})$. So there is a symmetry-breaking order parameter $(\theta + \tilde{\theta})$.

If $b, d = 0$ we have a similar scenario: $a + c \in 4\mathbb{Z}$ plus the locality constraint means both $a$ and $c$ are even. If $a = \pm c$ then there is just a single term $\cos[a(\phi \pm \tilde{\phi})]$, and we require $a \in 4\mathbb{Z}$. As we will show below, translation symmetry is broken by the order parameter $2(\phi + \tilde{\phi})$. For $a \neq \pm c$, we can combine the two arguments to form $(a + c)(\phi + \tilde{\phi})$, which again gives the order parameter $2(\phi + \tilde{\phi})$.

Details of the remaining cases are given in Appendix D. It follows that general symmetry allowed gapping terms always lead to spontaneous symmetry breaking. Thus the edge theory describes a non-trivial SPT phase.

## 5.5 The bulk-edge correspondence

Our low-energy edge theory is derived from a microscopic construction of the weak topological superconductor. The connection with the $\mathbb{Z}_4 \times \mathbb{Z}_2^{\mathsf{T}}$ FSPT phase has been somewhat implicit, only established through the general correspondence between topological phases with crystalline and internal symmetries discussed in Sec. 2.2. In this section we directly show that the edge theory captures correctly the anomaly expected for boundary states of the $\mathbb{Z}_4 \times \mathbb{Z}_2^{\mathsf{T}}$ FSPT phase. We provide two arguments for the bulk-edge correspondence. The arguments also provide evidence for stability of the gapless edge against the most general types of perturbations beyond those of null-vector type, since it is known that gapping terms which do not obey null-vector conditions can still open a gap [52, 53].

### 5.5.1 Domain wall structure

As reviewed in Sec 2, the ground state wavefunction of a $\mathbb{Z}_4 \times \mathbb{Z}_2^{\mathsf{T}}$ fermionic SPT phase can be understood using a decorated domain wall picture. While in the bulk domain walls are closed, they can terminate on the edge and a fermionic zero mode appears at the end point due to the decoration. This can be taken as a defining feature of the edge states: a $\mathbb{Z}_4$ domain wall binds a fermionic zero mode protected by the $\mathbb{Z}_2^{\mathsf{T}}$ symmetry.

We will now show that the edge theory does have the right domain wall structure. We first construct a gapping term which leads to spontaneous breaking of $T_t$ while preserving $T_r$. Consider a gapping term $U(\cos 4\phi + \cos 4\tilde{\phi})$, with $U < 0$, which condenses $\phi$ and $\tilde{\phi}$ at minima of the cosine potential $\frac{\pi m}{2}$ with $m \in \mathbb{Z}$. From the derived conditions on physical operators one can see that $2\phi$ and $2\tilde{\phi}$ are physical but $\phi$ and $\tilde{\phi}$ are not. The $T_t$ symmetry cycles through the ground state space

$$\begin{pmatrix} 2\phi \\ 2\tilde{\phi} \end{pmatrix}: \begin{pmatrix} 0 \\ 0 \end{pmatrix} \xrightarrow{T_t} \begin{pmatrix} \pi \\ 0 \end{pmatrix} \xrightarrow{T_t} \begin{pmatrix} \pi \\ \pi \end{pmatrix} \xrightarrow{T_t} \begin{pmatrix} 2\pi \\ \pi \end{pmatrix} \xrightarrow{T_t} \begin{pmatrix} 0 \\ 0 \end{pmatrix}, \tag{69}$$

while $T_r$ is unbroken.

Suppose we are in the state (denoted by $|0 \to \pi\rangle$) with a domain wall at $x$ separating the $\begin{pmatrix} 0 \\ 0 \end{pmatrix}$ state (denoted by $|0 \to 0\rangle$) and the $\begin{pmatrix} \pi \\ 0 \end{pmatrix}$ state. Note the following specific bosonic commutation relation

$$e^{\pm i \frac{\theta(y)}{2}} 2\phi(x) e^{\mp i \frac{\theta(y)}{2}} = 2\phi(x) + \begin{cases} \pm \pi & 0 < x < y \\ 0 & x > y \end{cases}. \tag{70}$$

We can create the domain wall configuration from a uniform ground state in two ways: two states

$$|0 \to \pi\rangle_{\pm} = e^{\pm i \frac{\theta(x)}{2}} |0 \to 0\rangle \tag{71}$$

are degenerate since they are related by the TR transformation. They have the same domain wall at $x$, separating the $\begin{pmatrix} 0 \\ 0 \end{pmatrix}$ and $\begin{pmatrix} \pi \\ 0 \end{pmatrix}$ states, but differ in local properties. Notice that while $e^{i\theta(x)/2}$ is non-local, in a closed system one always creates domain walls in pairs by applying $\exp\left[\frac{i}{2} \int_{x_0}^{x_1} \partial_x \theta \, dx\right]$, which is a physical string-like operator. If we look at the charge densities, $\rho_{\pm}(y) = \frac{1}{\pi} \langle 0 \to \pi | \partial_y \phi(y) | 0 \to \pi \rangle_{\pm}$, of the two states we see that

$$\rho_+(y) - \rho_-(y) = \delta(y - x). \tag{72}$$

.

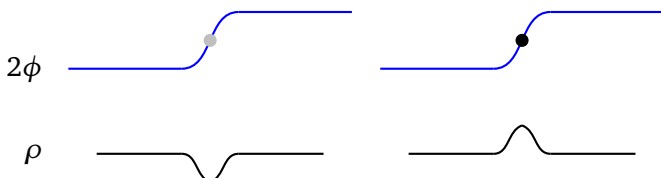

Figure 3: Degenerate domain wall states differ in their charge densities.

The degenerate kinked states differ in local charge $\Delta Q = 1$, suggesting the presence of a fermionic zero mode, as the only charge-1 local excitations in our system are physical fermions. In fact, an operator toggling between the two states is $e^{i(\phi + \tilde{\phi} + \theta)}$ (note that $\phi$ and $\tilde{\phi}$ condense). These two states are related by the TR transformation, protecting the degeneracy.

### 5.5.2 Gauging fermion parity

An alternative way to characterize the bulk SPT phase is through the symmetry properties of a fermion parity flux. In a nontrivial fermionic SPT phase, a fermion parity flux transforms projectively under the global symmetry group. In Ref. [54], a classification of 2D fermionic SPT phases was derived using these ideas. Mathematically, projective representation carried by a fermion parity flux is characterized by a 2-cocycle in $\mathcal{H}^2[G, \mathbb{Z}_2]$, which agrees with the group super-cohomology classification. We briefly summarize these facts about general classification of 2D FSPT phases in Appendix A.

We will directly couple the SPT phase to a $\mathbb{Z}_2$ gauge field, sourced by fermions. We will first carry out the gauging construction for the bulk theory. To this end, let us write down a topological field theory for the bulk:

$$\mathcal{L} = \sum_{IJ} \frac{K_{IJ}}{4\pi} a_I \wedge da_J + \cdots \tag{73}$$

Here $a_I$ are compact U(1) gauge fields, and the K matrix is given in Eq. (54). The same K matrix appears in the bulk Chern-Simons theory and the edge chiral boson theory following from the general bulk-boundary correspondence. $da$ is the fermion current in the bulk. Under symmetries, the gauge field transforms as:

$$T_g : a_I \rightarrow \sum_{IJ} (W_g)_{IJ} a_J, \tag{74}$$

where $W_g$ is given in Eqs. (55) and (56).

Now we couple the bulk to a $\mathbb{Z}_2$ gauge field $A$:

$$\frac{1}{2\pi}(a_1 + a_2 - a_3 - a_4)dA + \frac{1}{\pi}BdA. \tag{75}$$

The $\mathbb{Z}_2$ gauge theory is described by the mutual Chern-Simons term $\frac{1}{\pi}BdA$ (corresponding to a K matrix $\begin{pmatrix} 0 & 2 \\ 2 & 0 \end{pmatrix}$. $B$ can be thought of as a Higgs field that Higgs the U(1) gauge structure of $A$ down to $\mathbb{Z}_2$, and it couples to vortex current.

Here we choose $a_1 + a_2 - a_3 - a_4$ because this combination preserves $T_t$, and under $T_r$ it becomes minus itself. Therefore We let $T_t^{-1}AT_t = A$, $T_r^{-1}AT_r = A$, and $T_r^{-1}BT_r = -B$. We then integrate out $A$, which leads to a constraint $a_1 + a_2 - a_3 - a_4 + 2B = 0$. It can be resolved by

writing

$$
\begin{pmatrix} a_1 \\ a_2 \\ a_3 \\ a_4 \\ B \end{pmatrix} = \begin{pmatrix} 1 & 0 & 0 & 0 \\ -1 & 1 & 0 & 0 \\ 0 & 1 & 1 & 1 \\ 0 & 0 & -1 & 1 \\ 0 & 0 & 0 & -1 \end{pmatrix} \begin{pmatrix} \tilde{a}_1 \\ \tilde{a}_2 \\ \tilde{a}_3 \\ \tilde{a}_4 \end{pmatrix}. \tag{76}
$$

In fact, one can view the upper $4 \times 4$ block as the (non-invertible) similarity transformation between $a$ and $\tilde{a}$. We will denote it by $U$, with $\det U = 2$. In terms of the new variables $(\tilde{a}_1, \tilde{a}_2, \tilde{a}_3, \tilde{a}_4)$, the K matrix reads

$$
\begin{pmatrix} 0 & 1 & 0 & 0 \\ 1 & 0 & 1 & 1 \\ 0 & 1 & 0 & 2 \\ 0 & 1 & 2 & 0 \end{pmatrix}. \tag{77}
$$

This K matrix describes a $\mathbb{Z}_2$ topological order, as expected. Symmetry transformations are given by $\tilde{W}_g = U^{-1} W_g U, \delta \tilde{\phi}_g = U^{-1} \delta \phi_g$. The commutator between $T_t$ and $T_r$ acts on the corresponding edge fields as

$$
\tilde{\Phi} \to \tilde{\Phi} + \begin{pmatrix} 0 \\ 0 \\ \pi \\ \pi \end{pmatrix}. \tag{78}
$$

Notice that $e^{i\tilde{\phi}_3}$ are $e^{i\tilde{\phi}_4}$ are the two fermion parity fluxes, so they do transform projectively under $T_t$ and $T_r$, corresponding to the nontrivial cohomology class in $\mathcal{H}^2[\mathbb{Z}_4 \times \mathbb{Z}_2^\mathsf{T}, \mathbb{Z}_2]$. Physically, a fermion parity flux has a two-fold degeneracy protected by the symmetry.

# 6 Conclusion

In this work we find that edge modes of an intrinsically interacting FSPT phase can be described by a Luttinger liquid theory. It is possible that the same is true for all 2D FSPT phases in the group super-cohomology construction. We have proved that within $K$ matrix theory, $4 \times 4$ is the minimal dimension required for the $\mathbb{Z}_4 \times \mathbb{Z}_2^\mathsf{T}$ FSPT phase. An interesting question to ask is whether this $c = 2$ edge theory is the "minimal" (as measured by central charge) among all conformal field theories with the quantum anomaly. Another example of interacting 2D FSPT phase was found in Ref. [15], with symmetry group $\mathbb{Z}_4^f \times \mathbb{Z}_4 \times \mathbb{Z}_4$. Here the physics is somewhat different from the example discussed in this work: in the decorated domain wall construction, the state can be obtained by decorating $\mathbb{Z}_4$ domain walls with 1D $\mathbb{Z}_4^f \times \mathbb{Z}_4$ FSPT states. As mentioned in the introduction, the 1D FSPT phase itself can only exist in the presence of strong interactions, so is the 2D phase. It will be very interesting to construct a gapless edge theory for this interacting FSPT phase.

An important open problem is to understand the edge physics of intrinsically interacting FSPT phases beyond group super-cohomology [18, 19, 54]. An example of such phases in 2D arises with $\mathbb{Z}_8 \times \mathbb{Z}_2^\mathsf{T}$ symmetry. If the $\mathbb{Z}_8$ symmetry is replaced by translation, the bulk is a stack of Majorana chains, and the edge is a 1D chain with one Majorana per unit cell. The simplest Hamiltonian must involve four-site interactions. An example is the following Hamiltonian studied recently in Ref. [55]:

$$
H = g \sum_i \gamma_i \gamma_{i+1} \gamma_{i+3} \gamma_{i+4}. \tag{79}
$$

Remarkably, such a Hamiltonian is actually integrable [55], and realizes a gapless phase with a dynamical exponent $z = 3/2$. The nature of this phase is not fully understood. An interesting future direction is to construct other gapless theories, in particular conformal field theories, and develop field-theoretical descriptions.

Intrinsically interacting fermionic SPT phases also exist in three spatial dimensions [17–19]. Recent works have found general conditions on gapped surface topological order in the group super-cohomology cases [40, 56]. It will be interesting to explore gapless surface theories in these systems.

## Acknowledgements

J.S. acknowledges discussions with Aris Alexandradinata, Nick Bultinck, Judith Höller, Thomas Veness and Dominic Williamson. M.C. thanks Dominic Williamson and Chenjie Wang for conversations and collaborations on related topics, and Dave Aasen for pointing out a mistake in the draft.

**Funding information** MC acknowledges support from Alfred P. Sloan Foundation and the NSF under grant No. DMR-1846109.

## A   Group super-cohomology classification

Suppose the symmetry group is $G = \mathbb{Z}_2^f \times G_b$, where $G_b$ denotes the "bosonic" part of the symmetry group. In the group super-cohomology classification, 2D FSPT phases are labeled by a pair $(\nu, \omega)$ where $[\nu] \in \mathcal{H}^2[G_b, \mathbb{Z}_2]$ and $\omega \in C^3[G_b, U(1)]$. Here $[\cdot]$ denotes cohomology class. The $\mathbb{Z}_2$-valued 2-cocycle $\nu$'s are responsible for all the "intrinsically" FSPT phases. Note that $\nu$ needs to satisfy an obstruction-free condition, see Ref. [57] for a recent summary.

There are two ways to understand the physical meaning of $\nu$. First, in a decorated domain wall construction, domain walls are labeled by $\mathbf{g} \in G_b$. At a junction of three domain walls $\mathbf{g}, \mathbf{h}$ and $\mathbf{gh}$, one has a fermion mode whose occupation is determined by $\nu$. Denote $\mathbb{Z}_2$ additively as $\{0, 1\}$, then the occupation number is just $\nu(\mathbf{g}, \mathbf{h})$. Therefore $\nu$ determines the complex fermion decoration on domain wall junctions.

Alternatively, let us consider inserting a superconducting vortex into the FSPT phase. More precisely, such a vortex is a $\pi$ flux for fermions, i.e. a fermion picks up $-1$ phase factor when moving around the $\pi$ flux. Since the $\pi$ flux is not a local object, the symmetry group can be represented projectively on the flux. The projective representation, or more precisely the factor set, is given by $(-1)^\nu$.

Let us work this out for $G_b = \mathbb{Z}_4 \times \mathbb{Z}_2^\mathsf{T}$. The second cohomology group $\mathcal{H}^2[G_b, \mathbb{Z}_2] = \mathbb{Z}_2$. The nontrivial element of the $\mathbb{Z}_2$ corresponds to a two-dimensional projective representation, on which the generators of $\mathbb{Z}_4$ and $\mathbb{Z}_2^\mathsf{T}$ anticommute. In other words, the $\pi$ flux in the FSPT phase carries this projective representation.

## B   $2 \times 2$ K matrix

We show that a $2 \times 2$ K matrix can not describe the edge. For a non-chiral fermionic system, the K matrix can be fixed to be $K = \sigma^z$. Then it is straightforward to show that the only invertible similarity transformations that leave $\sigma^z$ invariant are $\mathbb{1}$ and $\sigma^z$. Similarly, the only ones that take $\sigma^z$ to $-\sigma^z$ are $\sigma^x$ and $\sigma^y$.

The time-reversal symmetry squaring to the identity is then implemented by $\sigma^x$. Because the $\mathbb{Z}_4$ generator $W_t$ has to commute with both $K$ and the time-reversal transformation, only $W_t = \mathbb{1}$ is allowed. At this point, notice that within $2 \times 2$ K matrix, the theory can be realized by free fermions.

We have found that the two symmetry transformations are given by

$$T_r : W_r = \sigma^x, \delta\phi_r = \begin{pmatrix} \alpha \\ -\alpha \end{pmatrix}, \tag{80}$$

and

$$T_t : W_t = \mathbb{1}, \delta\phi_t = \frac{\pi}{2}\begin{pmatrix} n_1 \\ n_2 \end{pmatrix}, n_{1,2} \in \{0, 1, 2, 3\}. \tag{81}$$

Further requiring $\mathbb{Z}_4$ commuting with $\mathsf{T}$ fixes $n_1 = n_2$.

With these transformations, the following perturbation is allowed $\cos(\phi_L - \phi_R - \alpha)$, which fully gaps out the edge without breaking symmetries.

## C    Symmetry actions on various operators

Let us work out how $T_r$ and $T_t$ act on the fermionic operators. From Eq. (12) we see

$$T_r : c_i \longrightarrow (-1)^i \prod_{j=1}^{i}(1 - 2\tilde{f}_j^\dagger \tilde{f}_j)f_i^\dagger = (-1)^i c_i^\dagger,$$

$$T_r : \tilde{c}_i \longrightarrow (-1)^{i-1} \prod_{j=1}^{i}(1 - 2f_j^\dagger f_j)\tilde{f}_i^\dagger = (-1)^{i-1}\tilde{c}_i^\dagger. \tag{82}$$

It should be noted also that $T_r : P_i \longrightarrow (-1)^N P_i$ . Under translation our physical fermions transform trivially as $f_i \to f_{i+1}$, adapting this to our partitioning of the even and odd sites gives $f_i \to \tilde{f}_{i+1}$ and $\tilde{f}_i \to f_i$ leading to

$$T_t : c_i \to \prod_{j=1}^{i}(1 - 2f_j^\dagger f_j)\tilde{f}_{j+1} = \tilde{c}_{i+1}$$

$$T_t : \tilde{c}_i \to \prod_{j=1}^{i-1}(1 - 2\tilde{f}_{j+1}^\dagger \tilde{f}_{j+1})f_i = c_i. \tag{83}$$

Recalling $c_k = \frac{1}{\sqrt{M}}\sum_{j=1}^{M} e^{-ikj}c_j$ we can use the transformation properties derived above to see what happens in the momentum basis.

$$T_r : c_k \longrightarrow \frac{1}{\sqrt{N}}\sum_{j=1}^{N} e^{ikj}(-1)^j c_j^\dagger = \frac{1}{\sqrt{N}}\sum_{j=1}^{N} e^{i(k+\pi)j}c_j^\dagger$$

$$= c_{k+\pi}^\dagger.$$

Similarly, $T_r : \tilde{c}_k \longrightarrow -\tilde{c}_{k+\pi}^\dagger$. For translation, using 83 we see

$$T_t : c_k \longrightarrow \frac{1}{\sqrt{N}}\sum_{j=1}^{N} e^{ikj}\tilde{c}_{j+1} = e^{-ik}\tilde{c}_k$$

$$T_t : \tilde{c}_k \longrightarrow \frac{1}{\sqrt{N}}\sum_{j=1}^{N} e^{ikj}c_j = c_k. \tag{84}$$

# D Perturbative stability of the edge theory

Here we give a proof via exhaustion that symmetry allowed Higgs terms push the edge theory away from a trivial phase. If we wish to gap out the system we must have exactly two linearly independent Higgs terms: $\delta\mathcal{L} = U_1\cos{(\mathbf{l}_1^T\Phi - \alpha_1)} + U_2\cos{(\mathbf{l}_2^T\Phi - \alpha_2)}$. We will focus on the $t > 0$ phase, the proof for the $t < 0$ phase follows in the same way. Recall how $\Phi$ transforms under $T_r$ and $T_t$:

$$T_t : \begin{pmatrix} \phi \\ \theta \\ \tilde{\phi} \\ \tilde{\theta} \end{pmatrix} \to \begin{pmatrix} \tilde{\phi} + \frac{\pi}{2} \\ \tilde{\theta} \\ \phi \\ \theta \end{pmatrix} \quad \text{and} \quad T_r : \begin{pmatrix} \phi \\ \theta \\ \tilde{\phi} \\ \tilde{\theta} \end{pmatrix} \to \begin{pmatrix} -\phi \\ \theta \\ -\tilde{\phi} \\ \tilde{\theta} + \pi \end{pmatrix} \tag{85}$$

a particular symmetry may act internally on each Higgs terms or it may exchange them. We will consider all the cases and show in each instance gapless modes or SSB are present.

**No exchange**

Consider the case in which each gapping term transforms trivially under both of the symmetries:

$$T_g^{-1}\cos{(\mathbf{1}^T\Phi - \alpha)}T_g = \cos{(\mathbf{1}^T\Phi - \alpha)}, g = t/r, \tag{86}$$

let $\mathbf{1}^T = (a, b, c, d)$, then acting with symmetry operators on $\cos{(a\phi + b\theta + c\tilde{\phi} + d\tilde{\theta} - \alpha)}$ we arrive at the following constraints:

| Symmetry | Vector constraint | Phase constraint |
|:---:|:---:|:---:|
| $T_t$ | $a = \pm c$ and $b = \pm d$ | $a \in 4\mathbb{Z}$ |
| $T_r$ | $a, c = 0$ or $b, d = 0$ | $d \in 2\mathbb{Z}$ |

If we explicitly consider some specific allowed term we can derive constraints on $\alpha$ (e.g something like $\cos{(2n(\theta + \tilde{\theta}) - \alpha)}$ is symmetry allowed for any $\alpha$ but $T_t$ constrains $\alpha$ in a term like $\cos{(2n(\theta - \tilde{\theta}) - \alpha)}$ to be $0, \pi$.) but this will not influence the presence of symmetry breaking behavior so we will ignore working these constraints out in general. From the table one has gapping terms of the form $\cos{(4n(\phi \pm \tilde{\phi}))}$ or $\cos{(2m(\theta \pm \tilde{\theta}))}$ which condense. Some fraction of these correspond to physical operators which break $T_t$ and $T_r$ respectively.

**$T_r$ exchange**

Now consider the case in which time reversal exchanges the two Higgs terms: explicitly this has the form

$$\delta L = U_1 [\cos{(a\phi + b\theta + c\tilde{\phi} + d\tilde{\theta} - \alpha)} \\ + (-1)^{d\pi}\cos{(-a\phi + b\theta - c\tilde{\phi} + d\tilde{\theta} - \alpha)}]. \tag{87}$$

Recall that if we are to condense the gapping terms the operators $\mathbf{l}_1^T\Phi, \mathbf{l}_2^T\Phi$ must commute with themselves and with each other. Suppose $T_t$ does not exchange the terms. One can check that the requirement that $\mathbf{l}_i K^{-1}\mathbf{l}_j = 0$ for $i, j = 1, 2$ is met only iff $a = 0$ or $b = 0$. We summarize the constraints in the following table

| Symmetry | Vector constraint | Phase constraint |
|:---:|:---:|:---:|
| $T_t$ | $a = \pm c$ and $b = \pm d$ | $a \in 4\mathbb{Z}$ |
| $\mathbf{l}_i^T K^{-1}\mathbf{l}_j = 0$ | $a = 0$ or $b = 0$ | |

Note now that $T_r^{-1}(\phi \pm \tilde{\phi})T_r \sim a(\phi \pm \tilde{\phi})$ and $T_r^{-1}(\theta \pm \tilde{\theta})T_r \sim b(\theta \pm \tilde{\theta})$. We are back to the previous case.

In the case where $T_t$ exchanges the Higgs terms we get:

| Symmetry | Vector constraint | Phase constraint |
|:---:|:---:|:---:|
| $T_t$ | $a = \pm c$ and $b = \mp d$ | $a \in 4\mathbb{Z}$ |
| $\mathbf{l}_i^T K^{-1} \mathbf{l}_j = 0$ | $a = 0$ or $b = 0$ | |

Again, the two gapping terms turn out to be proportional, and there is a symmetry breaking.

### $T_t$ exchanges

Finally, consider the last case in which $T_t$ exchanges the Higgs terms and $T_r$ does not. We have an interaction of the form $U_1[\cos(a\phi + b\theta + c\tilde{\phi} + d\tilde{\theta} - \alpha) + \cos(c\phi + d\theta + a\tilde{\phi} + b\tilde{\theta} + \frac{a\pi}{2} - \alpha)]$. There are only two Higgs terms so acting with $T_t$ twice must generate a phase of $2n\pi$.

| Symmetry | Vector constraint | Phase constraint |
|:---:|:---:|:---:|
| $T_t$ | | $a + c \in 4\mathbb{Z}$ |
| $T_r$ | $a, c = 0$ or $b, d = 0$ | $b, d \in 2\mathbb{Z}$ |

If $a, c = 0$ then $\delta L \sim \cos(2(n\theta + m\tilde{\theta}) - \alpha) + \cos(2(m\theta + n\tilde{\theta}) - \alpha)$ and there is either a gapless mode (if $n = \pm m$) or a physical fraction of $n\theta + m\tilde{\theta}, m\theta + n\tilde{\theta}$ and/or $(n + m)\theta + (m + n)\tilde{\theta}$ breaks symmetry.

If $b, d = 0$ we have a similar scenario: $a + c \in 4\mathbb{Z}$, together with the locality constraint we find that $a, c$ are both even, and we have the same situation as the $a, c = 0$ case treated just above.

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
