# Peer review of "Interacting Edge States of Fermionic Symmetry-Protected Topological Phases in Two Dimensions"

_SciPost Physics, doi:SciPost Phys. 9, 016 (2020)_

## Round 3 · Referee Report · Anonymous (Referee 1) · 2019-10-25

Strengths

Interesting new result.

Weaknesses

Poor presentation and incomplete references.

Report

Warnings issued while processing user-supplied markup:

  • Inconsistency: plain/Markdown and reStructuredText syntaxes are mixed. Markdown will be used.
    Add "#coerce:reST" or "#coerce:plain" as the first line of your text to force reStructuredText or no markup.
    You may also contact the helpdesk if the formatting is incorrect and you are unable to edit your text.

Summary of the manuscript:

The subject matter of this manuscript is the study of fermionic symmetry-protected topological (FSPT) phase of maters in two-dimensional space. The idea of this manuscript, as I understood it, is the following.

Consider first an open Kitaev chain, i.e., a one-dimensional superconductor that supports one Majorana zero mode at its left end and another one at its right end. Impose symmetry under complex conjugation so as to realize a fermionic symmetry protected topological (FSPT) phase in the symmetry class BDI.

Second, consider a pair of such Kitaev chains, stacked on top of each others. There are two Majorana zero modes on the left and two Majorana zero modes on the right. The left (right) pair of Majorana zero modes combines into a fermionic zero mode [the zero mode of a Schrieffer-Su-Heeger (SSH) chain] localized on the left (right) side of the SSH chain.

Third, stack infinitely many such SSH chains. Because of the symmetry under complex conjugation, there are infinitely many fermionic modes localized at the left end of each SSH chains. No one-body fermionic hopping between these fermionic zero modes is allowed by the BDI symmetry.

The idea of the authors is to write down a quartic interaction for the fermionic modes localized on the left sides of the SSH chains that is invariant under (i) the representation of the operation of complex conjugation for the original Majorana operators and (ii) any translation along the stacking direction. This fermionic model is an effectively one-dimensional lattice model that respects fermionic parity but does not conserve the fermion number, as it depends on a parameter $t\in\mathbb{R}$ compatible with the conservation of the fermion number and a parameter $\Delta\in\mathbb{R}$ that breaks the U(1) global gauge symmetry down to $\mathbb{Z}_{2}$.

This effective one-dimensional fermionic Hamiltonian is solved exactly by applying Jordan-Wigner transformations. It is found that there are gapful phases separated by gapless phases in the phase diagram. In the gapped phases, the representation of the operation of complex conjugation for the original Majorana operators is always broken. Moreover, the gapped phase with $t>0$ differs from the gapped phase with $t<0$ in that translation by one lattice spacing is also broken but not the product of the two broken symmetries when $t<0$. The gapful phases thus always break spontaneously at least one of the two protecting symmetries. These gapful phases are separated by quantum critical lines in parameter space. By the bulk-edge correspondence, these quantum critical lines are thus the signature of fermionic symmetry protected topological phases in two-dimensional space.

Finally, a stability analysis of the boundary theory is performed at the level of Abelian bosonization.

Criticism:

This manuscript is interesting and I believe that the results are solid. Unfortunately, it is poorly written. I fear that the first two sections in which the problem to be solved is motivated and the solution is explained are inaccessible to the target audience, at least it was inaccessible to this referee.

The bulk of the paper is accessible but written in a hurry, with little attention to details and precision. Citations are incomplete and bias.

Detailed comments:

Section 1:

The notation "\mathbb{Z}^{\mathsf{T}}_{2}'' is not explained. I am guessing that the superscript is supposed to imply that the physical interpretation of this group is that it is generated by reversal of time.

The sentence "As a result, a boundary without symmetry breaking is either gapless, or gapped with intrinsic topological order [4].'' only applies to a boundary whose dimensionality is larger than one.

"... namely intrinsically interacting fermionic SPT phases, ...'' should become "... namely intrinsically interacting fermionic SPT (FSPT) phases, ...''

"... can always be thought as a superposition of ...'' should become "... can always be thought of as a superposition of ...''

Section 2:

Section 2 is very hard to follow. This section reviews the concepts at work in this manuscript. However, it is written in such a compact way that I could only guess what the authors had in mind from reading the Sections in which calculations are presented.

I am not familiar with the statement "The ground state wavefunction of a SPT phase can always be thought as a superposition of all domain-wall configurations.''

I did not understand the sentence "Many nontrivial SPT phases can be constructed by decorating SPT states in one dimension lower on domain walls.''

From the context, it seems that "\mathbb{Z}^{\mathsf{T}}_{2}'' refers to a reversal of time that squares to unity. Why not explain this? According to the tenfold way, the symmetry classes CII and DIII are also protected by time-reversal symmetry, but one that squares to minus unity. I am not sure how the authors would denote such a symmetry, but the very existence of the symmetry classes CII and DIII caused my confusion.

The sentence "In this work we adopt a different approach, invoking the connection between SPT phases with internal symmetry and those with crystalline symmetry of the same group structure. This correspondence has been formalized in Ref. [31], and verified in many examples.'' is written in a way to discourage any reader except perhaps authors of Ref. 31. More pedagogy would be welcome in this paragraph.

The last paragraph of Section 2 is again written as if one needed to belong to a secret brotherhood to understand what is meant.

Section 3:

Could the authors clarify if the particle-hole transformation defined in Eq. (3) is antiunitary. It should be since it originates from complex conjugation, but it would be nice to remove any ambiguity since particle-hole transformations of operators are often defined to be unitary.

The range $i=1,...,2N$ should be declared in Eq. (2), not only in Fig.\ 1.

The definition of the "physical fermion'' deserves its numbered equation together with the algebra that it obeys.

Why is the fermion operator that is constructed from a pair of Majorana operators called "physical'' after Eq. (2)?

The sentence "We are interested in symmetric phases of the BDI chain, which is necessarily gapless.'' is confusing for two reasons. First, an explanation would be needed. Naively, one might invoke the Lieb-Schultz-Mattis (LSM) theorem, but this theorem presumes a U(1) symmetry, which is broken in Eq. (4). Second, I thought that a fair amount of space is also devoted to the gaped phases in this manuscript.

It is not specified if $t$ and $\Delta$ are real valued or complex valued in Eq. (4). Both options are possible to the best of my knowledge at this stage, but I believe that the authors restrict themselves to $t$ and $\Delta$ real valued later on.

The statement that the pair of fermion operators defined in Eq. (9) commute should come just after (9), not after (10). Moreover, \qquad should be inserted between the definitions of $c_n$ and $\tilde{c}_{n}$. [Same applies to Eq. (8).]

The range of $i$ in Eq. (10) is not the same as that of $i$ in Eq.\ (7). The authors could have reserved the letter $j,n=1,...,N$ when distinguishing even from odd sites as they did in Eqs. (8) and (9). The same comment applies to Eqs.\ (11), (12).

Section 4:

The main goal of this Section is to show that the gaped phases of the boundary Hamiltonian break spontaneously some of its symmetries. It would be helpful to announce what will be done in this Section in the first paragraph of this Section.

I do not understand the sentence "Since the model is supposed to describe an anomalous edge, the ground state is either gapless, or gapped with spontaneous breaking of the symmetries.'' The dichotomy between gaplessness or a gap with spontaneous symmetry breaking (SSB) seems to originate from the one-dimensional nature of the boundary, not from it supporting an anomalous theory. As I said earlier, a reference explaining this dichotomy would be nice if it exists (LSM theorem cannot be applied).

I do not understand this claim "..., which are gapped whenever $\Delta\neq 0$.'' What happens when $t=0$, is there a gap?

Equation (17) seems to demand that $\Delta$ and $t$ are real valued.

Equation (18) needs some horizontal spaces for readability. Equation (18) should have been presented after Eq.\ (7).

Equation (22) is incomplete as it is not specified explicitly of which Pauli matrix $\boldsymbol{\tau}_{i}$ $|\uparrow\rangle_{i}$ is eigenstate with eigenvalue +1. If the action of $T_r$ on $|\uparrow\rangle_{i}$ had been given explicitly, it would be obvious by inspection that eigenstates (23) and (25) are not eigenstates of $T_r$.

Section 5:

Section 5 deals with the line of critical points $\Delta=0$. When $\Delta=0$, the boundary Hamiltonian is gapless. This is a necessary condition for the two-dimensional theory to realize a fermionic symmetry protected boundary. To show that the $\Delta=0$ condition is sufficient for the two-dimensional theory to realize a fermionic symmetry protected boundary, the authors will prove that this line of quantum critical points is robust to interactions that are allowed by the protecting symmetries. To this end, they will describe the line of critical points $\Delta=0$ by a bosonic theory and use and a reasoning going back to a seminal paper by Haldane to show that no interaction can gap this bosonic theory without spontaneously breaking the protecting symmetries. This bosonic theory is referred to as a "chiral bosonic theory''. I will come back to this terminology below.

Equation (30) omits the sector with $\tilde{c}$ operators. This seems to be a typo.

"... do form the right group structure faithfully.'' should be changed to "... do represent faithfully the group $\mathrm{Z}^{\mathsf{T}}_{2}\times\mathbb{Z}$.''

The references in the sentence "Here we discuss an alternative formulation using K matrix [39–42], ...'' are incomplete and bias. The second paper after Ref.\ [39] to be cited in this context should be that of Haldane 10.1103/PhysRevLett.74.2090, since it is in this paper that the condition (54) was introduced for the first time, to the best of my knowledge. Haldane as Wen were only considering $K$ matrices with a net chirality (their traces was non vanishing). Early manuscripts dealing with the stability of the boundary theory using non-chiral bosonization techniques ($K$ matrices whose traces are vanishing) are those of Levin and Stern, 10.1103/PhysRevLett.103.196803, and Neupert et al., 10.1103/PhysRevB.84.165107. The extension to all ten symmetry classes from the tenfold way was done by Neupert et al in 10.1103/PhysRevB.90.205101.

I do not understand the sentence "... to understand the perturbative stability.''. My understanding of Haldane's 10.1103/PhysRevLett.74.2090 is that his approach to the stability of the edge states was designed to be nonperturbative, as opposed to that used by Kane, Fisher, and Polchinski in 10.1103/PhysRevLett.72.4129.

The references [42,43] in the sentence "In an effort to gap out the theory we can consider adding Higgs terms of the form [42, 43] ...'' are problematic for two reasons. First, Ref.\ [43] is the same as Ref.\ [40]. Second both References are written after Neupert et al., 10.1103/PhysRevB.84.165107 in which the strategy of "adding Higgs terms '' is applied to study the effects of strong interactions on the boundary of a topological insulator in the symmetry class AII.

I have the same difficulty as above with the use of "perturbatively'' in "Through a null vector analysis, we have shown that perturbatively the gapless edge modes are protected by the symmetry.''

I would like to comment about the use of the adjective "chiral''. The authors correctly use the adjective chiral when describing operators that obey either left-moving or right-moving equations of motion. The authors state correctly that their K matrix is "non chiral'' as it is traceless. I am not sure it is wise to call Eq. (42) a general chiral Luttinger liquid if one does not state explicitly that the K matrix is not traceless. For this reason, I would omit chiral in the sentence "... and the edge chiral boson theory following ...'' and in the first section of Section 6. To put it differently, I can always recast a nonchiral bosonic theory in one dimension in terms of pairs of chiral fields with opposite chiralities, but a bosonic theory with an odd number of chiral fields is necessarily chiral.

Section 6:

The case $t=0$ was never discussed.

It is reiterated by the authors that the main result of this manuscript is the construction of a FSPT phase in two-dimensional space that cannot be realized by free fermions. In this context, I would like to point out that a (nonchiral) FSPT phase that cannot be realized with free fermions was constructed by Neupert et al.\ 10.1103/PhysRevB.90.205101. This FSPT phase displays a nonvanishing quantum Hall conductivity with vanishing thermal Hall conductivity. I do not know if this was the first example of a FSPT phase that has no free fermion counterpart, but it is certainly one of the early examples.

Requested changes

See Report

  • validity: good
  • significance: good
  • originality: good
  • clarity: low
  • formatting: below threshold
  • grammar: acceptable

Author:  Meng Cheng  on 2020-05-27  [id 839]

(in reply to Report 1 on 2019-10-25)
Category:
remark
answer to question

We are very grateful for the careful review by the referee and have revised the manuscript accordingly. Below we address comments in the report about the scientific content of the work.

Sec. 2

As suggested by the report, we have significantly expanded Sec. 2 to give a more pedagogical introduction of the background concepts used in this work. In particular, two new subsections were added, one on the decorated domain wall construction, the other on the correspondence between SPT phases protected by crystalline and internal symmetries.

Sec. 3

R: The sentence "We are interested in symmetric phases of the BDI chain, which is necessarily gapless.'' is confusing for two reasons. First, an explanation would be needed. Naively, one might invoke the Lieb-Schultz-Mattis (LSM) theorem, but this theorem presumes a U(1) symmetry, which is broken in Eq. (4). Second, I thought that a fair amount of space is also devoted to the gaped phases in this manuscript.

The referee is right that the LSM can not be invoked here since pairing is allowed. The gaplessness is protected by both translation and time-reversal symmetries. While this is generally expected from bulk-edge correspondence, it will become clearer when an edge theory is available as the later sections unfold, so we decided to remove this sentence to avoid the confusion.

Sec. 4

R: I do not understand the sentence "Since the model is supposed to describe an anomalous edge, the ground state is either gapless, or gapped with spontaneous breaking of the symmetries.'' The dichotomy between gaplessness or a gap with spontaneous symmetry breaking (SSB) seems to originate from the one-dimensional nature of the boundary, not from it supporting an anomalous theory. As I said earlier, a reference explaining this dichotomy
would be nice if it exists (LSM theorem cannot be applied).

The dichotomy follows from both the anomalous nature of the boundary and the one dimensionality. The anomaly forbids a trivial non-degenerate ground state, so generally there are three possibilities: gapless, spontaneous symmetry breaking or a symmetric topological order. However the last option is not available in one dimension, so we are left with the first two.

Q: I do not understand this claim "..., which are gapped whenever $\Delta\neq 0$''. What happens when t=0, is there a gap?

The referee is correct that the model is gapped if and only if both $t$ and $\Delta$ are nonzero. We have revised the manuscript accordingly.

Sec. 5

R: The references in the sentence "Here we discuss an alternative formulation using K matrix [39–42], ...'' are incomplete and bias. The second paper after Ref.\ [39] to be cited in this context should be that of Haldane 10.1103/PhysRevLett.74.2090, since it is in this paper that the condition (54) was introduced for the first time, to the best of my knowledge. Haldane as Wen were only considering K matrices with a net chirality (their traces was non vanishing). Early manuscripts dealing with the stability of the boundary theory using non-chiral bosonization techniques
(K matrices whose traces are vanishing) are those of Levin and Stern, 10.1103/PhysRevLett.103.196803, and Neupert et al., 10.1103/PhysRevB.84.165107. The extension to all ten symmetry classes from the tenfold way was done by Neupert et al in 10.1103/PhysRevB.90.205101.

We agree with the comment of the referee and have revised the citations accordingly.

R: I do not understand the sentence "... to understand the perturbative stability.''. My understanding of Haldane's 10.1103/PhysRevLett.74.2090 is that his approach to the stability of the edge states was designed to be nonperturbative, as opposed to that used by Kane, Fisher, and Polchinski in 10.1103/PhysRevLett.72.4129.

Here by "perturbative stability" we had in mind gapping perturbations of null-vector type, i.e. the approach first discussed in Haldane's work 10.1103/PhysRevLett.74.2090 as cited by the referee. However, it is also known that edge theories can be gapped by perturbations not of null-vector type, an example of which was given in Ref. [51] (of the current manuscript). We agree with the referee that this could be confusing, and have rewritten the opening paragraph of Sec. 5.4 to clarify what is being done in that section.

R: I have the same difficulty as above with the use of "perturbatively'' in "Through a null vector analysis, we have shown that perturbatively the gapless edge modes are protected by the symmetry.''

We have changed the title of the section to "The bulk-edge correspondence", as that is what the actual content. We also revised the comment about stability, removing "perturbative" or "non-perturbative" and stating more explicitly that the stability should be addressed beyond null-vector type gapping terms.

R: The references [42,43] in the sentence "In an effort to gap out the theory we can consider adding Higgs terms of the form [42, 43] ...'' are problematic for two reasons. First, Ref.\ [43] is the same as Ref.\ [40]. Second both References are written after Neupert et al., 10.1103/PhysRevB.84.165107 in which the strategy of "adding Higgs terms '' is applied to study the effects of strong interactions on the boundary of a topological insulator in the symmetry class AII.

We thank the referee for catching the typo. The references have been updated.

R: I would like to comment about the use of the adjective "chiral''. The authors correctly use the adjective chiral when describing operators that obey either left-moving or right-moving equations of motion. The authors state correctly that their K matrix is "non chiral'' as it is traceless. I am not sure it is wise to call Eq. (42) a general chiral Luttinger liquid if one does not state explicitly that the K matrix is not traceless. For this reason, I would omit chiral in the sentence "... and the edge chiral boson theory following ...'' and in the first section of Section 6. To put it differently, I can always recast a nonchiral bosonic theory in one dimension in terms of pairs of chiral fields with opposite chiralities,
but a bosonic theory with an odd number of chiral fields is necessarily chiral.

We thank the referee for the comment, and it is indeed true that our edge theory is non-chiral. We have changed the sentence to avoid confusion.

Sec. 6

R: The case t=0 was never discussed.

We restrict our attention to t>0, as stated in the beginning of Sec. 5.

R: It is reiterated by the authors that the main result of this manuscript is the construction of a FSPT phase in two-dimensional space that cannot be realized by free fermions. In this context, I would like to point out that a (nonchiral) FSPT phase that cannot be realized with free fermions was constructed by Neupert et al.\ 10.1103/PhysRevB.90.205101.
This FSPT phase displays a nonvanishing quantum Hall conductivity with vanishing thermal Hall conductivity. I do not know if this was the first example of a FSPT phase that has no free fermion counterpart, but it is certainly one of the early examples.

Indeed the state constructed in Neupert et al.\ 10.1103/PhysRevB.90.205101 is an example of FSPT phase with no free analog. However, this phase can be understood as the integer quantum Hall state of charge-2e bosons. In this work we are more interested in interacting FSPT phases which have no bosonic analog. We have expanded the introduction to include examples of this type.

---

## Round 3 · Referee Report · Anonymous (Referee 2) · 2019-11-4

Report

The question studied by the authors, namely possible phases of edge mode of a topological phase due to interactions on the edge, is interesting and timely. The example that they work out is valuable and is written out clearly (Sections 3-5). Unfortunately, the introduction (Sections 1-2) is not well written, with many terms being left undefined ($Z_2^T$, FSPT, to state two examples), and many assumptions on prior knowledge of the reader being made. I propose that the authors make an effort to clarify the presentation in these sections. Once this is done, I recommend the paper for publication.

---

## Round 5 · Referee Report · Anonymous (Referee 2) · 2020-6-30

Report

I join the other referee in recommending publication.

---

## Round 5 · List of Changes

Major changes:

  1. We have significantly expanded section 2 to include more background materials on concepts and ideas used in this work.

  2. We renamed Section 5.5 to better reflect the actual content of the section.

  3. Citations on gapping of null-vector type in a general Luttinger liquid are updated.

---

## Editorial Decision

published